# Kernelized information bottleneck leads to biologically plausible 3-factor Hebbian learning in deep networks

**Roman Pogodin**
Gatsby Computational Neuroscience Unit
University College London
London, W1T 4JG
`roman.pogodin.17@ucl.ac.uk`

**Peter E. Latham**
Gatsby Computational Neuroscience Unit
University College London
London, W1T 4JG
`pel@gatsby.ucl.ac.uk`

## Abstract

The state-of-the art machine learning approach to training deep neural networks, backpropagation, is implausible for real neural networks: neurons need to know their outgoing weights; training alternates between a bottom-up forward pass (computation) and a top-down backward pass (learning); and the algorithm often needs precise labels of many data points. Biologically plausible approximations to backpropagation, such as feedback alignment, solve the weight transport problem, but not the other two. Thus, fully biologically plausible learning rules have so far remained elusive. Here we present a family of learning rules that does not suffer from any of these problems. It is motivated by the information bottleneck principle (extended with kernel methods), in which networks learn to compress the input as much as possible without sacrificing prediction of the output. The resulting rules have a 3-factor Hebbian structure: they require pre- and post-synaptic firing rates and an error signal – the third factor – consisting of a global teaching signal and a layer-specific term, both available without a top-down pass. They do not require precise labels; instead, they rely on the similarity between pairs of desired outputs. Moreover, to obtain good performance on hard problems and retain biological plausibility, our rules need divisive normalization – a known feature of biological networks. Finally, simulations show that our rules perform nearly as well as backpropagation on image classification tasks.

## 1 Introduction

Supervised learning in deep networks is typically done using the backpropagation algorithm (or backprop), but in its present form it cannot explain learning in the brain [1]. There are three reasons for this: weight updates require neurons to know their *outgoing* weights, which they do not (the weight transport problem); the forward pass for computation and the backward pass for weight updates need separate pathways and have to happen sequentially (preventing updates of earlier layers before the error is propagated back from the top ones, see Fig. 1A); and a large amount of precisely labeled data is needed.

While approximations to backprop such as feedback alignment [2, 3] can solve the weight transport problem, they do not eliminate the requirement for a backward pass or the need for labels. There have been suggestions that a backward pass could be implemented with apical dendrites [4], but it's not clear how well the approach scales to large networks, and the backward pass still has to follow the forward pass in time.

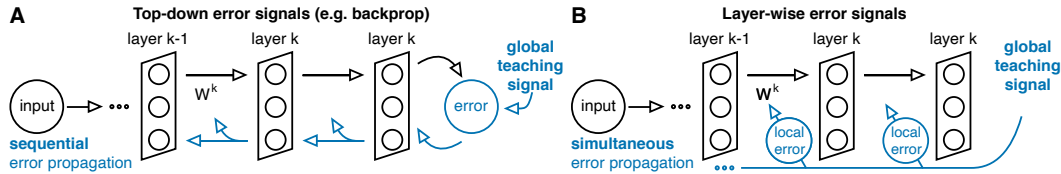

Figure 1: **A.** The global error signal is propagated to each layer from the layer above, and used to update the weights. **B.** The global error signal is sent directly to each layer.

Backprop is not, however, the only way to train deep feedforward networks. An alternative is to use so-called layer-wise update rules, which require only activity in adjacent (and thus connected) layers, along with a global error signal (Fig. 1B). Layer-wise training removes the need for both weight transport and a backward pass, and there is growing evidence that such an approach can work as well as backprop [5, 6, 7]. However, while such learning rules are local in the sense that they mainly require activity only in adjacent layers, that does not automatically imply biological plausibility.

Our work focuses on finding a layer-wise learning rule that is biologically plausible. For that we take inspiration from the information bottleneck principle [8, 9], in which every layer minimizes the mutual information between its own activity and the input to the network, while maximizing the mutual information between the activity and the correct output (e.g., a label). Estimating the mutual information is hard [10], so [11] proposed the HSIC bottleneck: instead of mutual information they used "kernelized cross-covariance" called Hilbert-Schmidt independence criterion (HSIC). HSIC was originally proposed in as a way to measure independence between distributions [12]. Unlike mutual information, HSIC is easy to estimate from data [12], and the information bottleneck objective keeps its intuitive interpretation. Moreover, as we will see, for classification with roughly balanced classes it needs only pairwise similarities between labels, which results in a binary teaching signal.

Here we use HSIC, but to achieve biologically plausible learning rules we modify it in two ways: we replace the HSIC between the input and activity with the kernelized covariance, and we approximate HSIC with "plausible HSIC", or pHSIC, the latter so that neurons don't need to remember their activity over many data points. (However, the objective function becomes an upper bound to the HSIC objective.) The resulting learning rules have a 3-factor Hebbian structure: the updates are proportional to the pre- and post-synaptic activity, and are modulated by a third factor (which could be a neuromodulator [13]) specific to each layer. In addition, to work on hard problems and remain biologically plausible, our update rules need divisive normalization, a computation done by the primary visual cortex and beyond [14, 15].

In experiments we show that plausible rules generated by pHSIC work nearly as well as backprop on MNIST [16], fashion-MNIST [17], Kuzushiji-MNIST [18] and CIFAR10 [19] datasets. This significantly improves the results from the original HSIC bottleneck paper [11].

## 2   Related work

Biologically plausible approximations to backprop solve the weight transport problem in multiple ways. Feedback alignment (FA) [3] and direct feedback alignment (DFA) [20] use random fixed weights for the backward pass but scale poorly to hard tasks such as CIFAR10 and ImageNet [21, 22], However, training the feedback pathway to match the forward weights can achieve backprop-level performance [2]. The sign symmetry method [23] uses the signs of the feedforward weights for feedback and therefore doesn't completely eliminate the weight transport problem, but it scales much better than FA and DFA [2, 24]. Other methods include target prop [25, 26] (scales worse than FA [22]) and equilibrium prop [27] (only works on simple tasks, but can work on CIFAR10 at the expense of a more complicated learning scheme [28]). However, these approaches still need to alternate between forward and backward passes.

To avoid alternating forward and backward passes, layer-wise objectives can be used. A common approach is layer-wise classification: [29] used fixed readout weights in each layer (leading to slightly worse performance than backprop); [6] achieved backprop-like performance with trainable readout weights on CIFAR10 and CIFAR100; and [5] achieved backprop-like performance on ImageNet with multiple readout layers. However, layer-wise classification needs precise labels and local backprop (or

its approximations) for training. Methods such as contrastive learning [7] and information [9] or HSIC [11] bottleneck and gated linear networks [30, 31] provide alternatives to layer-wise classification, but don't focus on biological plausibility. Biologically plausible alternatives with weaker supervision include similarity matching [6, 32], with [6] reporting a backprop-comparable performance using cosine similarity, and fully unsupervised rules such as [33, 34]. Our method is related to similarity matching; see below for additional discussion.

## 3 Training deep networks with layer-wise objectives: a kernel methods approach and its plausible approximation

Consider an $L$-layer feedforward network with input $\mathbf{x}$, layer activity $\mathbf{z}^k$ (for now without divisive normalization) and output $\hat{\mathbf{y}}$,

$$\mathbf{z}^1 = f\left(\mathbf{W}^1 \mathbf{x}\right), \ldots, \mathbf{z}^L = f\left(\mathbf{W}^L \mathbf{z}^{L-1}\right); \; \hat{\mathbf{y}} = f\left(\mathbf{W}^{L+1} \mathbf{z}^L\right). \tag{1}$$

The standard training approach is to minimize a loss, $l(\mathbf{y}, \hat{\mathbf{y}})$, with respect to the weights, where $\mathbf{y}$ is the desired output and $\hat{\mathbf{y}}$ is the prediction of the network. Here, though, we take an alternative approach: we use layer-wise objective functions, $l_k(\mathbf{x}, \mathbf{z}^k, \mathbf{y})$ in layer $k$, and minimize each $l_k(\mathbf{x}, \mathbf{z}^k, \mathbf{y})$ with respect to the weight in that layer, $\mathbf{W}^k$ (simultaneously for every $k$). The performance of the network is still measured with respect to $l(\mathbf{y}, \hat{\mathbf{y}})$, but that quantity is explicitly minimized only with respect to the output weights, $\mathbf{W}^{L+1}$.

To choose the layer-wise objective function, we turn to the information bottleneck [8], which minimizes the information between the input and the activity in layer $k$, while maximizing the information between the activity in layer $k$ and the desired output [9]. Information, however, is notoriously hard to compute [10], and so [11] proposed an alternative based on the Hilbert-Schmidt Independence Criterion (HSIC) – the HSIC bottleneck. HSIC is a kernel-based method for measuring independence between probability distribution [12]. Similarly to the information bottleneck, this method tries to balance compression of the input with prediction of the correct output, with a (positive) balance parameter $\gamma$,

$$\min_{\mathbf{W}^k} \mathrm{HSIC}\left(X, Z^k\right) - \gamma\, \mathrm{HSIC}\left(Y, Z^k\right), \; k = 1, \ldots, L, \tag{2}$$

where $X, Z^k$ and $Y$ are random variables, with a distribution induced by the input (the $\mathbf{x}$) and output (the $\mathbf{y}$). HSIC is a measure of dependence: it is zero if its arguments are independent, and increases as dependence increases,

$$\mathrm{HSIC}\left(A, B\right) = \int \Delta P_{ab}(\mathbf{a}_1, \mathbf{b}_1)\, k(\mathbf{a}_1, \mathbf{a}_2)\, k(\mathbf{b}_1, \mathbf{b}_2)\, \Delta P_{ab}(\mathbf{a}_2, \mathbf{b}_2), \tag{3}$$

where

$$\Delta P_{ab}(\mathbf{a}, \mathbf{b}) \equiv \left(p_{ab}(\mathbf{a}, \mathbf{b}) - p_a(\mathbf{a})p_b(\mathbf{b})\right) d\mathbf{a}\, d\mathbf{b}. \tag{4}$$

The kernels, $k(\cdot, \cdot)$ (which might be different for $\mathbf{a}$ and $\mathbf{b}$), are symmetric and positive definite functions, the latter to insure that HSIC is non-negative. More details on kernels and HSIC are given in Appendix A.

HSIC gives us a layer-wise cost function, which eliminates the need for backprop. However, there is a downside: estimating it from data requires memory. This becomes clear when we consider the empirical estimator of Eq. (3) given $m$ observations [12],

$$\widehat{\mathrm{HSIC}}(A, B) = \frac{1}{m^2} \sum_{ij} k(\mathbf{a}_i, \mathbf{a}_j) k(\mathbf{b}_i, \mathbf{b}_j) + \frac{1}{m^2} \sum_{ij} k(\mathbf{a}_i, \mathbf{a}_j) \frac{1}{m^2} \sum_{kl} k(\mathbf{b}_k, \mathbf{b}_l)$$
$$- \frac{2}{m^3} \sum_{ijk} k(\mathbf{a}_i, \mathbf{a}_k) k(\mathbf{b}_j, \mathbf{b}_k). \tag{5}$$

In a realistic network, data points are seen one at a time, so to compute the right hand side from $m$ samples, $m - 1$ data point would have to be remembered. We solve this in the usual way, by stochastic gradient descent. For the first term we use two data points that are adjacent in time; for

the second, we accumulate and store the average over the kernels (see Eq. (11) below). The third term, however, is problematic; to compute it, we would have to use three data points. Because this is implausible, we make the approximation

$$\frac{1}{m}\sum_k k(\mathbf{a}_i, \mathbf{a}_k)k(\mathbf{b}_j, \mathbf{b}_k) \approx \frac{1}{m^2}\sum_{kl} k(\mathbf{a}_i, \mathbf{a}_k)k(\mathbf{b}_j, \mathbf{b}_l). \tag{6}$$

Essentially, we replace the third term in Eq. (5) with the second. This leads to "plausible" HSIC, which we call pHSIC,

$$\text{pHSIC}(A,\ B) = (\mathbb{E}_{\mathbf{a}_1\mathbf{b}_1}\mathbb{E}_{\mathbf{a}_2\mathbf{b}_2} - \mathbb{E}_{\mathbf{a}_1}\mathbb{E}_{\mathbf{b}_1}\mathbb{E}_{\mathbf{a}_2}\mathbb{E}_{\mathbf{b}_2})(k(\mathbf{a}_1,\mathbf{a}_2)k(\mathbf{b}_1,\mathbf{b}_2)). \tag{7}$$

While pHSIC is easier to compute than HSIC, there is still a potential problem: computing HSIC $(X, Z^k)$ requires $k(\mathbf{x}_i, \mathbf{x}_j)$, as can be seen in the above equation. But if we knew how to build a kernel that gives a reasonable distance between inputs, we wouldn't have to train the network for classification. So we make one more change: rather than minimizing the dependence between $X$ and $Z^k$ (by minimizing the first term in Eq. (2)), we minimize the (kernelized) covariance of $Z^k$. To do this, we replace $X$ with $Z^k$ in Eq. (2), and define pHSIC $(A,\ A)$ via Eq. (7) but with $p_{ab}(\mathbf{a}, \mathbf{b})$ set to $p_a(\mathbf{a})\delta(\mathbf{a} - \mathbf{b})$,

$$\text{pHSIC}(A,\ A) = \mathbb{E}_{\mathbf{a}_1}\mathbb{E}_{\mathbf{a}_2}(k(\mathbf{a}_1,\mathbf{a}_2))^2 - (\mathbb{E}_{\mathbf{a}_1}\mathbb{E}_{\mathbf{a}_2}k(\mathbf{a}_1,\mathbf{a}_2))^2. \tag{8}$$

This gives us the new objective,

$$\min_{\mathbf{W}^k} \text{pHSIC}(Z^k,\ Z^k) - \gamma\,\text{pHSIC}(Y,\ Z^k),\ k = 1, \ldots, L. \tag{9}$$

The new objective preserves the intuition behind the information bottleneck, which is to throw away as much information as possible about the input. It's also an upper bound on the "true" HSIC objective as long as the kernel over the desired outputs is centered ($\mathbb{E}_{\mathbf{y}_1}k(\mathbf{y}_1, \mathbf{y}_2) = 0$; see Appendix A.2), and doesn't significantly change the performance (see Appendix D).

Centering the output kernel is straightforward in classification with balanced classes: take $\mathbf{y}$ to be centered one-hot encoded labels (for $n$ classes, $y_i = 1 - 1/n$ for label $i$ and $-1/n$ otherwise), and use the cosine similarity kernel $k(\mathbf{y}_1, \mathbf{y}_2) = \mathbf{y}_1^\top \mathbf{y}_2/\|\mathbf{y}_1\|\|\mathbf{y}_2\|$. In addition, the teaching signal is binary in this case: $k(\mathbf{y}_i, \mathbf{y}_j) = 1$ if $\mathbf{y}_i = \mathbf{y}_j$ and $-1/(n-1)$ otherwise. We will use exactly this signal in our experiments, as the datasets we used are balanced. For slightly unbalanced classes, the signal can still be binary (see Appendix A.3).

When $\gamma = 2$, our objective is related to similarity matching. For the cosine similarity kernel over activity, it is close to [6], and for the linear kernel, it is close to [32, 33]. However, the update rule for the cosine similarity kernel is implausible, and for the linear kernel the rule performs poorly (see below and in Appendix B). We thus turn to the Gaussian kernel.

## 4 Circuit-level details of the gradient: a hidden 3-factor Hebbian structure

### 4.1 General update rule

To derive the update rule for gradient descent, we need to estimate the gradient of Eq. (9). This is relatively straightforward, so we leave the derivation to Appendix B, and just report the update rule,

$$\Delta\mathbf{W}^k \propto \sum_{ij}\left(\gamma\mathring{k}(\mathbf{y}_i, \mathbf{y}_j) - 2\mathring{k}(\mathbf{z}_i^k, \mathbf{z}_j^k)\right)\frac{d}{d\mathbf{W}^k}k(\mathbf{z}_i^k, \mathbf{z}_j^k), \tag{10}$$

where the circle above $k$ means empirical centering,

$$\mathring{k}(\mathbf{a}_i, \mathbf{a}_j) \equiv k(\mathbf{a}_i, \mathbf{a}_j) - \frac{1}{m^2}\sum_{i'j'} k(\mathbf{a}_{i'}, \mathbf{a}_{j'}). \tag{11}$$

This rule has a 3-factor form with a global and a local part (Fig. 2A): for every pair of points $i$ and $j$, every synapse needs the same multiplier, and this multiplier needs the similarities between labels (the global signal) and layer activities (the local signal) on two data points. As mentioned in the previous section, on our problems the teaching signal $k(\mathbf{y}_i, \mathbf{y}_j)$ is binary, but we will keep the more general notation here.

However, the derivative in Eq. (10) is not obviously Hebbian. In fact, it gives rise to a simple Hebbian update only for some kernels. Below we consider the Gaussian kernel, and in Appendix B.4 we show that the cosine similarity kernel (used in [6]) produces an unrealistic rule.

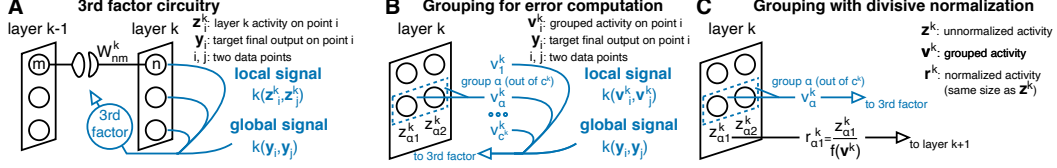

Figure 2: **A.** The weight update uses a 3rd factor consisting of a global teaching signal and a local (to the layer) signal, both capturing similarities between two data points. **B.** Grouping: neurons in layer $k$ form $c^k$ groups, each represented by a single number; those numbers are used to compute the local signal. **C.** Grouping with divisive normalization: the activity of every neuron is normalized using the grouped signal before passing to the next layer (the error is computed as in B).

### 4.2 Gaussian kernel: two-point update

The Gaussian kernel is given by $k(\mathbf{z}_i^k, \mathbf{z}_j^k) = \exp(-\|\mathbf{z}_i^k - \mathbf{z}_j^k\|^2/2\sigma^2)$. To compute updates from a stream of data (rather than batches), we approximate the sum in Eq. (10) with only two points, for which we'll again use $i$ and $j$. If we take a linear fully connected layer (see Appendix B.2 for the general case), the update over two points is

$$\Delta \mathbf{W}^k \propto M_{ij}^k (\mathbf{z}_i^k - \mathbf{z}_j^k)(\mathbf{z}_i^{k-1} - \mathbf{z}_j^{k-1})^\top \tag{12a}$$

$$M_{ij}^k = -\frac{1}{\sigma^2}\left(\gamma \, \overset{\circ}{k}(\mathbf{y}_i, \mathbf{y}_j) - 2 \, \overset{\circ}{k}(\mathbf{z}_i^k, \mathbf{z}_j^k)\right) k(\mathbf{z}_i^k, \mathbf{z}_j^k). \tag{12b}$$

Here $M_{ij}^k$ is the layer-specific third factor.

The role of the third factor is to ensure that if labels, $\mathbf{y}_i$ and $\mathbf{y}_j$, are similar, then the activity, $\mathbf{z}_i^k$ and $\mathbf{z}_j^k$, is also similar, and vice-versa. To see why it has that effect, assume $\mathbf{y}_i$ and $\mathbf{y}_j$ are similar and $\mathbf{z}_i^k$ and $\mathbf{z}_j^k$ are not. That makes $M_{ij}^k$ negative, so the update rule is anti-Hebbian, which tends to move activity closer together. Similarly, if $\mathbf{y}_i$ and $\mathbf{y}_j$ are not similar and $\mathbf{z}_i^k$ and $\mathbf{z}_j^k$ are, $M_{ij}^k$ is positive, and the update rule is Hebbian, which tends to move activity farther apart.

### 4.3 Gaussian kernel with grouping and divisive normalization

The Gaussian kernel described above works well on small problems (as we'll see below), but in wide networks it needs to compare very high-dimensional vectors, for which there is no easy metric. To circumvent this problem, [6] computed the variance of the response over each channel in each convolutional layers, and then used it for cosine similarity.

We will use the same approach, which starts by arranging neurons into $c^k$ groups, labeled by $\alpha$, so that $z_{\alpha n}^k$ is the response of neuron $n$ in group $\alpha$ of layer $k$ (Fig. 2B). We'll characterize each group by its "smoothed" variance (meaning we add a positive offset $\delta$), denoted $u_\alpha^k$,

$$u_\alpha^k \equiv \frac{\delta}{c_\alpha^k} + \frac{1}{c_\alpha^k}\sum_{n'}\left(\overset{\circ}{z}_{\alpha n'}^k\right)^2 ; \quad \overset{\circ}{z}_{\alpha n}^k \equiv z_{\alpha n}^k - \frac{1}{c_\alpha^k}\sum_{n'} z_{\alpha n'}^k, \tag{13}$$

where $c_\alpha^k$ is the number of neurons in group $\alpha$ of layer $k$. One possible kernel would compare the standard deviation (so the squared root of $u_\alpha^k$) across different data points. However, we can get better performance by exponentiation and centering across channels. We thus define a new variable $v_\alpha^k$,

$$v_\alpha^k = (u_\alpha^k)^{1-p} - \frac{1}{c^k}\sum_{\alpha'}(u_{\alpha'}^k)^{1-p}, \tag{14}$$

and use this grouped activity in the Gaussian kernel,

$$k(\mathbf{z}_i^k, \mathbf{z}_j^k) = \exp\left(-\frac{1}{2\sigma^2}\left\|\mathbf{v}_i^k - \mathbf{v}_j^k\right\|^2\right), \tag{15}$$

where, recall, $i$ and $j$ refer to different data points, and $(\mathbf{v}^k)_\alpha = v_\alpha^k$.

To illustrate the approach, we consider a linear network; see Appendix B for the general case. Taking the derivative of $k(\mathbf{z}_i^k, \mathbf{z}_j^k)$ with respect to the weight in layer $k$, we arrive at

$$\frac{d\,k(\mathbf{z}_i^k, \mathbf{z}_j^k)}{d\,W_{\alpha nm}^k} \propto -\frac{1}{\sigma^2} k(\mathbf{z}_i^k, \mathbf{z}_j^k)\,(v_{\alpha,\,i}^k - v_{\alpha,\,j}^k) \left( \frac{\overset{\circ k}{z}_{\alpha n,\,j}}{(u_{\alpha,\,i}^k)^p} z_{m,\,i}^{k-1} - \frac{\overset{\circ k}{z}_{\alpha n,\,j}}{(u_{\alpha,\,j}^k)^p} z_{m,\,j}^{k-1} \right) . \tag{16}$$

Because of the term $u_\alpha^k$ in this expression, the learning rule is no longer strictly Hebbian. We can, though, make it Hebbian by assuming that the activity of the presynaptic neurons is $\overset{\circ k}{z}_{\alpha n,\,j}/(u_{\alpha,\,j}^k)^p$,

$$z_{\alpha n}^k = f\left( \sum_m W_{\alpha nm}^k r_m^{k-1} \right); \quad r_{\alpha n}^k = \frac{\overset{\circ k}{z}_{\alpha n}}{(u_\alpha^k)^p} . \tag{17}$$

This automatically introduces divisive normalization into the network (Fig. 2C), a common "canonical computation" in the brain [14], and in our case makes the update 3-factor Hebbian. It also changes the network from Eq. (1) to Eq. (17), but that turns out to improve performance. The resulting update rule (again for a linear network; see Appendix B.3 for the general case) is

$$\Delta W_{\alpha nm}^k \propto M_{\alpha,\,ij}^k \left( r_{\alpha n,\,i}^k r_{m,\,i}^{k-1} - r_{\alpha n,\,j}^k r_{m,\,j}^{k-1} \right)$$
$$M_{\alpha,\,ij}^k = -\frac{1}{\sigma^2} \left( \gamma \overset{\circ}{k}(\mathbf{y}_i, \mathbf{y}_j) - 2\,\overset{\circ}{k}(\mathbf{z}_i^k, \mathbf{z}_j^k) \right) k(\mathbf{z}_i^k, \mathbf{z}_j^k)\,(v_{\alpha,\,i}^k - v_{\alpha,\,j}^k) . \tag{18}$$

The circuitry to implement divisive normalization would be recurrent, but is out of scope of this work (see, however, [35] for network models of divisive normalization). For a convolutional layer, $\alpha$ would denote channels (or groups of channels), and the weights would be summed within each channel for weight sharing. When $p$, the exponent in Eq. (14), is equal to 0.5, our normalization scheme is equivalent to divisive normalization in [36] and to group normalization [37].

Note that the rule is slightly different from the one for the basic Gaussian kernel (Eq. (12)): the weight change is no longer proportional to differences of pre- and post-synaptic activities; instead, it is proportional to the difference in their product times the global factor for the channel. This form bears a superficial resemblance to Contrastive Hebbian learning [38, 39]; however, that method doesn't have a third factor, and it generates points $i$ and $j$ using backprop-like feedback connections.

### 4.4 Online update rules for the Gaussian kernel are standard Hebbian updates

Our objective and update rules so far have used batches of data. However, we introduced pHSIC in Section 3 because a realistic network has to process data point by point. To show how this can work with our update rules, we first switch indices of points $i,\ j$ to time points $t,\ t - \Delta t$.

**Gaussian kernel**

The update in Eq. (12) becomes

$$\Delta \mathbf{W}^k(t) \propto M_{t,\,t-\Delta t}^k (\mathbf{z}_t^k - \mathbf{z}_{t-\Delta t}^k)(\mathbf{z}_t^{k-1} - \mathbf{z}_{t-\Delta t}^{k-1})^\top . \tag{19}$$

As an aside, if the point $t - \Delta t$ was presented for some period of time, its activity can be replaced by the mean activity: $\mathbf{z}_{t-\Delta t}^k \approx \boldsymbol{\mu}_t^k$ and $z_{t-\Delta t}^{k-1} \approx \boldsymbol{\mu}_t^{k-1}$. The update becomes a standard Hebbian one,

$$\Delta \mathbf{W}^k(t) \propto M_{t,\,t-\Delta t}^k (\mathbf{z}_t^k - \boldsymbol{\mu}_t^k)(\mathbf{z}_t^{k-1} - \boldsymbol{\mu}_t^{k-1})^\top . \tag{20}$$

**Gaussian kernel with divisive normalization**

The update in Eq. (18) allows two interpretations. The first one works just like before: we first introduce time, and then assume that the previous point $r_{\alpha n,\,t-\Delta t}^k$ is close to the short-term average of activity $\mu_{\alpha n,\,t}^k$. This results in

$$\Delta W_{\alpha nm}^k(t) \propto M_{\alpha,\,t,\,t-\Delta t}^k \left( r_{\alpha n,\,t}^k r_{m,\,t}^{k-1} - \mu_{\alpha n,\,t}^k \mu_{m,\,t}^{k-1} \right) . \tag{21}$$

The second one uses the fact that for points at times $t - \Delta t,\ t,\ t + \Delta t$ the Hebbian term of point $t$ appears twice: first as $M_{\alpha,\,t,\,t-\Delta t}^k r_{\alpha n,\,t}^k r_{m,\,t}^{k-1}$, and then as $-M_{\alpha,\,t+\Delta t,\,t}^k r_{\alpha n,\,t}^k r_{m,\,t}^{k-1}$. Therefore we can separate the Hebbian term at time $t$ and write the update as

$$\Delta W_{\alpha nm}^k(t) \propto \left( M_{\alpha,\,t,\,t-\Delta t}^k - M_{\alpha,\,t+\Delta t,\,t}^k \right) r_{\alpha n,\,t}^k r_{m,\,t}^{k-1} . \tag{22}$$

While the Hebbian part of Eq. (22) is easier than in Eq. (21), it requires the third factor to span a longer time period. In both cases (and also for the plain Gaussian kernel), computing the third factor with local circuitry is relatively straightforward; for details see Appendix C.

## 5 Experiments

### 5.1 Experimental setup

We compared our learning rule against stochastic gradient descent (SGD), and with an adaptive optimizer and batch normalization. While networks with adaptive optimizers (e.g. Adam [40]) and batch normalization (or batchnorm, [41]) perform better on deep learning tasks and are often used for biologically plausible algorithms (e.g. [2, 6]), these features imply non-trivial circuitry (e.g. the need for gradient steps in batchnorm). As our method focuses on what circuitry implements learning, the results on SGD match this focus better.

We used the batch version of the update rule (Eq. (10)) only to make large-scale simulations computationally feasible. We considered the Gaussian kernel, and also the cosine similarity kernel, the latter to compare with previous work [6]. (Note, however, that the cosine similarity kernel gives implausible update rules, see Appendix B.) For both kernels we tested 2 variants of the rules: plain (without grouping), and with grouping and divisive normalization (as in Eq. (18)). (Grouping *without* divisive normalization performed as well or worse, and we don't report it here as the resulting update rules are less plausible; see Appendix D.) We also tested backprop, and learning of only the output layer in small-scale experiments to have a baseline result (also with divisive normalization in the network). In large-scale experiments, we also compared our approach to feedback alignment [3], sign symmetry [23] and layer-wise classification [29, 6].

The nonlinearity was leaky ReLU (LReLU [42]; with a slope of $0.1$ for the negative input), but for convolutional networks trained with SGD we changed it to SELU [43] as it performed better. All parameters (learning rates, learning rate schedules, grouping and kernel parameters) were tuned on a validation set (10% of the training set). Optimizing HSIC instead of our approximation, pHSIC, didn't improve performance, and the original formulation of the HSIC bottleneck (Eq. (2)) performed much worse (not shown). The datasets were MNIST [16], fashion-MNIST [17], Kuzushiji-MNIST [18] and CIFAR10 [19]. We used standard data augmentation for CIFAR10 [19], but no augmentation for the other datasets. All simulation parameters are provided in Appendix D. The implementation is available on GitHub: https://github.com/romanpogodin/plausible-kernelized-bottleneck.

### 5.2 Small fully connected network

We start with a 3-layer fully connected network (1024 neurons in each layer). To determine candidates for large-scale experiments, we compare the proposed rules to each other and to backprop. We thus delay comparison with other plausible learning methods to the next section; for performance of plausible methods in shallow networks see e.g. [44]. The models were trained with SGD for 100 epochs with dropout [45] of $0.05$, batch size of 256, and the bottleneck balance parameter $\gamma = 2$ (other values of $\gamma$ performed worse); other parameters are provided in Appendix D.

Our results (summarized in Table 1 for mean test accuracy; see Appendix D for deviations between max and min) show a few clear trends across all four datasets: the kernel-based approaches with grouping and divisive normalization perform similarly to backprop on easy datasets (MNIST and its slightly harder analogues), but not on CIFAR10; grouping and divisive normalization had little effect on the cosine similarity performance; the Gaussian kernel required grouping and divisive normalization for decent accuracy. However, we'll see that the poor performance on CIFAR10 is not fundamental to the method; it's because the network is too small.

### 5.3 Large convolutional networks and CIFAR10

Because all learning rules perform reasonably well on MNIST and its related extensions, in what follows we consider only CIFAR10, with the architecture used in [6]: **conv128-256-maxpool-256-512-maxpool-512-maxpool-512-maxpool-fc1024** and its version with double the convolutional channels (denoted 2x; the size of the fully connected layer remained the same). All networks were

trained for 500 epochs with $\gamma = 2$, dropout of $0.05$ and batch size of 128 (accuracy jumps in Fig. 3 indicate learning rate decreases); the rest of the parameters are provided in Appendix D.

Table 1: Mean test accuracy over 5 runs for a 3-layer (1024 neurons each) fully connected net. Last layer: training of the last layer; cossim: cosine similarity; grp: grouping; div: divisive normalization.

| | backprop | | last layer | | pHSIC: cossim | | pHSIC: Gaussian | |
|---|---|---|---|---|---|---|---|---|
| | | div | | div | | grp+div | | grp+div |
| MNIST | 98.6 | 98.4 | 92.0 | 95.4 | 94.9 | 96.3 | 94.6 | 98.1 |
| fashion-MNIST | 90.2 | 90.8 | 83.3 | 85.7 | 86.3 | 88.1 | 86.5 | 88.8 |
| Kuzushiji-MNIST | 93.4 | 93.5 | 71.2 | 78.2 | 80.4 | 87.2 | 80.2 | 91.1 |
| CIFAR10 | 60.0 | 60.3 | 39.2 | 38.0 | 51.1 | 47.6 | 41.4 | 46.4 |

Table 2: Mean test accuracy on CIFAR10 over 5 runs for the 7-layer conv nets (1x and 2x wide). FA: feedback alignment; sign sym.: sign symmetry; layer class.: layer-wise classification; cossim: cosine similarity; divnorm: divisive normalization; bn: batchnorm.

| | backprop | FA | sign sym. | layer class. | | pHSIC + grouping | |
|---|---|---|---|---|---|---|---|
| | | | | | +FA | cossim | Gaussian |
| 1x + SGD + divnorm | 91.0 | 80.4 | 89.5 | 90.5 | 81.0 | 89.8 | 86.2 |
| 2x + SGD + divnorm | 90.9 | 80.6 | 91.3 | 91.3 | 81.2 | 91.3 | 90.4 |
| 1x + AdamW + bn | 94.1 | 82.4 | 93.6 | 92.1 | 90.3 | 91.3 | 89.9 |
| 2x + AdamW + bn | 94.3 | 81.6 | 93.9 | 92.1 | 91.1 | 91.9 | 91.0 |

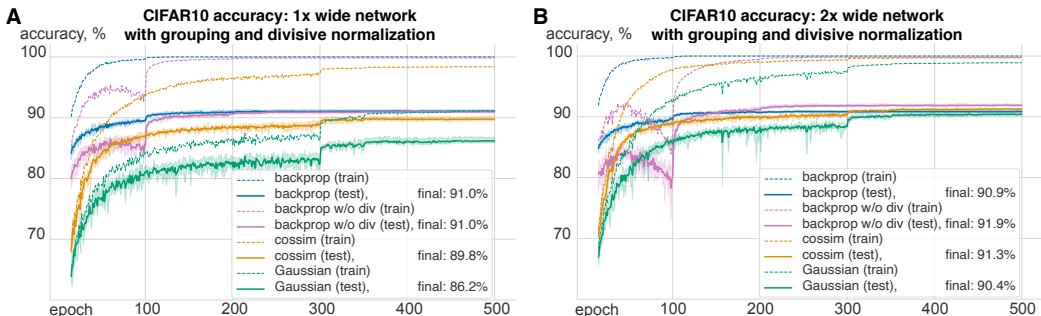

Figure 3: Performance of backprop, cosine similarity kernel (cossim) and Gaussian kernel on CIFAR10 with SGD, grouping and divisive normalization (and without for backprop; in pink). Solid lines: mean test accuracy over 5 random seeds; shaded areas: min/max test accuracy over 5 seeds; dashed lines: mean training accuracy over 5 seeds. **A.** 1x wide network: cosine similarity nearly matches backprop but doesn't achieve perfect training accuracy; Gaussian kernel lags behind cosine similarity. **B.** 2x wide network: backprop performance slightly improves; both kernels nearly match backprop performance, but Gaussian kernel still doesn't achieve perfect training accuracy.

For SGD with divisive normalization (and also grouping for pHSIC-based methods; first two rows in Table 2), layer-wise classification, sign symmetry and the cosine similarity kernel performed as well as backprop (consistent with previous findings [6, 24]); feedback alignment (and layer-wise classification with FA) performed significantly worse. In those cases, increasing the width of the network had a marginal effect on performance. The Gaussian kernel performed worse than backprop on the 1x network, but increasing the width closed the performance gap. A closer look at the learning dynamics of pHSIC-based methods and backprop (Fig. 3) reveals low training accuracy with the Gaussian kernel on the 1x wide net, vs. almost 100% for 2x, explaining low test accuracy.

For AdamW [46] with batchnorm [41] (last two rows in Table 2), performance improved for all objectives, but it improved more for backprop (and sign symmetry) than for the pHSIC-based

objectives. However, batch normalization (and in part adaptive learning rates of AdamW) introduces yet another implausible feature to the training method due to the batch-wide activity renormalization.

Only backprop, layer-wise classification (without feedback alignment) and the cosine similarity kernels performed well without any normalization (see Appendix D); for the other methods some kind of normalization was crucial for convergence. Backprop without divisive normalization (solid pink line in Fig. 3) had non-monotonic performance, which can be fixed with a smaller learning rate at the cost of slightly worse performance.

# 6 Discussion

We proposed a layer-wise objective for training deep feedforward networks based on the kernelized information bottleneck, and showed that it can lead to biologically plausible 3-factor Hebbian learning. Our rules works nearly as well as backpropagation on a variety of image classification tasks. Unlike in classic Hebbian learning, where the pre- and post-synaptic activity triggers weight changes, our rules suggest large *fluctuations* in this activity should trigger plasticity (e.g. when a new object appears).

Our learning rules do not need precise labels; instead they need only a binary signal: whether or not the previous and the current point have the same label. This allows networks to build representations with weaker supervision. We did train the last layer with precise labels, but that was only to compute accuracy; the network would learn just as well without it. To completely avoid supervision in hidden layers, it is possible to adapt our learning scheme to the contrastive (or self-supervised) setting as in Contrastive Predictive Coding [47, 7] and SimCLR [48].

Our rules do, though, need a global signal, which makes up part of the third factor. Where could it come from? In the case of the visual system, we can think of it as a "teaching" signal coming from other sensory areas. For instance, the more genetically pre-defined olfactory system might tell the brain that two successively presented object smell differently, and therefore should belong to different classes. The third factor also contains a term that is local to the layer, but requires averaging of activity within the layer. This could be done by cells that influence plasticity, such as dopaminergic neurons [13] or glia [49].

Although our approach is a step towards fully biologically plausible learning rules, it still suffers from some unrealistic features. The recurrence in our networks is limited to that necessary for divisive normalization, and has no excitatory within-layer recurrence or top-down signals. Those might be necessary to go from image classification to more realistic tasks (e.g., video). Our networks also allow negative firing rates due to the use of leaky ReLU and SELU nonlinearities. The latter (which we used to compensate for the lack of batchnorm) saturates for large negative inputs, and therefore the activity of each neuron can be viewed as a value relative to the background firing. Our main experiments also use convolutional networks, which are implausible due to weight sharing among neurons. Achieving good performance without weight sharing is an open question, although there are some results for backprop [22].

We showed that our rules can compete with backprop and its plausible approximations on CIFAR10, even though they rely on less supervision and simpler error signals. It should be possible to scale our learning rules to larger datasets, such as CIFAR100 [19] and ImageNet [50], as suggested by results from other layer-wise rules [5, 6, 7]. The layer-wise objectives can also make the theoretical analysis of deep learning easier. In fact, recent work analyzed a similar type of kernel-based objectives, showing its optimality with one hidden layer and backprop-comparable performance in deeper networks [51].

The human brain contains about $10^{11}$ neurons, of which only about $10^6$ – less than one per 100,000 – are directly connected to the outside world; the rest make up hidden layers. Understanding how such a system updates its synaptic strengths is one of the most challenging problems in neuroscience. We proposed a family of biologically plausible learning rules for feedforward networks that have the potential to solve this problem. For a complete understanding of learning they need, of course, to be adapted to unsupervised and recurrent settings, and verified experimentally. In addition, our learning rules are much more suitable for neuromorphic chips than standard backprop, due to the distributed nature of weight updates, and so could massively improve their scalability.

## Broader Impact

This research program, like most in neuroscience, has the potential to advance our understanding of the brain. This comes with a host of societal implications. On the upside, it can give us a deeper understanding of mental health, possibly providing new therapies – something that would improve the lives of on the order of 1 billion people. On the downside, a deeper understanding of the brain is likely to translate into accelerated development of artificial intelligence, which would put a great deal of power into the hands of a small number of people.

## Acknowledgments

This work was supported by the Gatsby Charitable Foundation and the Wellcome Trust.

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
