[Supplementary Material]

# Appendices

## A Kernel methods, HSIC and pHSIC

### A.1 Kernels and HSIC

A kernel is a symmetric function $k(\mathbf{a}_1, \mathbf{a}_2)$ that maps $\mathbb{R}^n \times \mathbb{R}^n \to \mathbb{R}$ and is positive-definite,

$$\forall \mathbf{a}_i \in \mathbb{R}^n, \forall c_i \in \mathbb{R}, \quad \sum_{ij} c_i c_j k(\mathbf{a}_i, \mathbf{a}_j) \geq 0 \,. \tag{23}$$

Consequently, the matrix $K_{ij} = k(\mathbf{a}_i, \mathbf{a}_j)$ is also positive-semidefinite.

We will use the following kernels,

$$\text{linear:} \qquad k(\mathbf{a}_i, \mathbf{a}_j) = \mathbf{a}_i^\top \mathbf{a}_j \,; \tag{24}$$

$$\text{cosine similarity:} \qquad k(\mathbf{a}_i, \mathbf{a}_j) = \mathbf{a}_i^\top \mathbf{a}_j / (\|\mathbf{a}_i\|_2 \|\mathbf{a}_j\|_2) \,; \tag{25}$$

$$\text{Gaussian:} \qquad k(\mathbf{a}_i, \mathbf{a}_j) = \exp(-\|\mathbf{a}_i - \mathbf{a}_j\|_2^2 / (2\sigma^2)) \,. \tag{26}$$

The Hilbert-Schmidt Independence Criterion (introduced in [12]) measures independence between two random variable, $A$ and $B$, with kernels $k_a$ and $k_b$,

$$\text{HSIC}(A, B) = (\mathbb{E}_{\mathbf{a}_1 \mathbf{b}_1} \mathbb{E}_{\mathbf{a}_2 \mathbf{b}_2} - 2\, \mathbb{E}_{\mathbf{a}_1 \mathbf{b}_1} \mathbb{E}_{\mathbf{a}_2} \mathbb{E}_{\mathbf{b}_2} + \mathbb{E}_{\mathbf{a}_1} \mathbb{E}_{\mathbf{b}_1} \mathbb{E}_{\mathbf{a}_2} \mathbb{E}_{\mathbf{b}_2}) (k(\mathbf{a}_1, \mathbf{a}_2)\, k(\mathbf{b}_1, \mathbf{b}_2)) \,, \tag{27}$$

where $\mathbb{E}_{\mathbf{a}_1 \mathbf{b}_1}$ denotes expectation over the joint distribution.

When $B \equiv A$, meaning that $p_{ab}(\mathbf{a}, \mathbf{b}) = p_a(\mathbf{a})\, \delta(\mathbf{b} - \mathbf{a})$, HSIC becomes

$$\text{HSIC}(A, A) = \mathbb{E}_{\mathbf{a}_1} \mathbb{E}_{\mathbf{a}_2} (k(\mathbf{a}_1, \mathbf{a}_2))^2 - 2\, \mathbb{E}_{\mathbf{a}_1} (\mathbb{E}_{\mathbf{a}_2} k(\mathbf{a}_1, \mathbf{a}_2))^2 + (\mathbb{E}_{\mathbf{a}_1} \mathbb{E}_{\mathbf{a}_2} k(\mathbf{a}_1, \mathbf{a}_2))^2 \,. \tag{28}$$

If both kernels are linear, it is easy to show that HSIC becomes the squared Frobenius norm of the cross-covariance,

$$\text{HSIC}(A, B) = \|\mathbf{C}_{\mathbf{ab}}\|_F^2 \,, \quad \mathbf{C}_{\mathbf{ab}} = \mathbb{E}_{\mathbf{ab}}\, \mathbf{a}\mathbf{b}^\top \,. \tag{29}$$

In general, HSIC follows the same intuition – it is the squared Hilbert-Schmidt norm (generalization of the Frobenius norm) of the cross-covariance operator.

### A.2 pHSIC

We define the "plausible" HSIC by substituting $\mathbb{E}_{\mathbf{a}_1 \mathbf{b}_1} \mathbb{E}_{\mathbf{a}_2} \mathbb{E}_{\mathbf{b}_2}$ in Eq. (27) by $\mathbb{E}_{\mathbf{a}_1} \mathbb{E}_{\mathbf{b}_1} \mathbb{E}_{\mathbf{a}_2} \mathbb{E}_{\mathbf{b}_2}$,

$$\text{pHSIC}(A, B) = (\mathbb{E}_{\mathbf{a}_1 \mathbf{b}_1} \mathbb{E}_{\mathbf{a}_2 \mathbf{b}_2} - \mathbb{E}_{\mathbf{a}_1} \mathbb{E}_{\mathbf{b}_1} \mathbb{E}_{\mathbf{a}_2} \mathbb{E}_{\mathbf{b}_2}) (k(\mathbf{a}_1, \mathbf{a}_2)\, k(\mathbf{b}_1, \mathbf{b}_2)) \,. \tag{30}$$

Therefore, $\text{HSIC}(A, B) = \text{pHSIC}(A, B)$ when $\mathbf{a}$ and $\mathbf{b}$ are independent ($\mathbb{E}_{\mathbf{a}_1 \mathbf{b}_1} = \mathbb{E}_{\mathbf{a}_1} \mathbb{E}_{\mathbf{b}_1}$, which is not useful in our case), and when $\mathbb{E}_{\mathbf{a}_2} k(\mathbf{a}_1, \mathbf{a}_2) = 0$ or $\mathbb{E}_{\mathbf{b}_2} k(\mathbf{b}_1, \mathbf{b}_2) = 0$.

In addition, $\text{pHSIC}(A, A)$ becomes the variance of $k(\mathbf{a}_1, \mathbf{a}_2)$ with respect to $p_a(\mathbf{a}_1) p_a(\mathbf{a}_2)$,

$$\text{pHSIC}(A, A) = \mathbb{E}_{\mathbf{a}_1} \mathbb{E}_{\mathbf{a}_2} (k(\mathbf{a}_1, \mathbf{a}_2))^2 - (\mathbb{E}_{\mathbf{a}_1} \mathbb{E}_{\mathbf{a}_2} k(\mathbf{a}_1, \mathbf{a}_2))^2 = \mathbb{V}\text{ar}(k(\mathbf{a}_1, \mathbf{a}_2)) \,. \tag{31}$$

By combining Eq. (28) and Eq. (31), we can show that $\text{HSIC}(A, A) \leq \text{pHSIC}(A, A)$: denoting $\mu_a(\mathbf{a}_1) = \mathbb{E}_{\mathbf{a}_2} k(\mathbf{a}_1, \mathbf{a}_2)$, we have

$$\begin{aligned} \text{pHSIC}(A, A) - \text{HSIC}(A, A) &= 2\, \mathbb{E}_{\mathbf{a}_1} (\mathbb{E}_{\mathbf{a}_2} k(\mathbf{a}_1, \mathbf{a}_2))^2 - 2\, (\mathbb{E}_{\mathbf{a}_1} \mathbb{E}_{\mathbf{a}_2} k(\mathbf{a}_1, \mathbf{a}_2))^2 \\ &= 2\left( \mathbb{E}_{\mathbf{a}_1} \mu_a(\mathbf{a}_1)^2 - (\mathbb{E}_{\mathbf{a}_1} \mu_a(\mathbf{a}_1))^2 \right) \\ &= 2\, \mathbb{V}\text{ar}(\mu_a(\mathbf{a}_1)) \geq 0. \end{aligned} \tag{32}$$

As a result, our objective,

$$\text{pHSIC}\left(Z^k, Z^k\right) - \gamma\, \text{pHSIC}\left(Y, Z^k\right) \,, \tag{33}$$

is an upper bound on the "true" objective whenever $\text{pHSIC}\left(Y, Z^k\right) = \text{HSIC}\left(Y, Z^k\right)$, which is the case when $\mathbb{E}_{\mathbf{y}_2} k(\mathbf{y}_1, \mathbf{y}_2) = 0$.

Finally, the empirical estimate of pHSIC (which can be derived just like for HSIC in [12]) is

$$\widehat{\text{pHSIC}}(A, B) = \frac{1}{m^2} \sum_{ij} k(\mathbf{a}_i, \mathbf{a}_j) k(\mathbf{b}_i, \mathbf{b}_j) - \frac{1}{m^2} \sum_{ij} k(\mathbf{a}_i, \mathbf{a}_j) \frac{1}{m^2} \sum_{ql} k(\mathbf{b}_q, \mathbf{b}_l) \,. \tag{34}$$

### A.3 How much information about the label do we need?

In our rules, the information about the label comes only through $k(\mathbf{y}_i, \mathbf{y}_j)$, where $\mathbf{y}$ is a one-hot vector – for $n$ classes, an $n$-dimensional vector of mainly zeros with only a single one (which corresponds to its label). For $k(\mathbf{y}_i, \mathbf{y}_j)$ we use the cosine similarity kernel, with $\mathbf{y}$ centered. If the dataset is balanced (i.e., all classes have the same probability, $1/n$), $\mathbb{E}_{\mathbf{y}_j} k(\mathbf{y}_i, \mathbf{y}_j) = 0$ and the resulting kernel is

$$k(\mathbf{y}_i, \mathbf{y}_j) = \frac{(\mathbf{y}_i - \frac{1}{n}\mathbf{1}_n)^\top (\mathbf{y}_j - \frac{1}{n}\mathbf{1}_n)}{\left\|\mathbf{y}_i - \frac{1}{n}\mathbf{1}_n\right\| \left\|\mathbf{y}_j - \frac{1}{n}\mathbf{1}_n\right\|} = \frac{\mathbb{I}\left[\mathbf{y}_i = \mathbf{y}_j\right] - \frac{1}{n}}{1 - \frac{1}{n}} = \begin{cases} 1, & \mathbf{y}_i = \mathbf{y}_j, \\ -\frac{1}{n-1}, & \text{otherwise}. \end{cases} \tag{35}$$

If there are many classes, this signal approaches $\mathbb{I}\left[\mathbf{y}_i = \mathbf{y}_j\right]$, which is what the same as the uncentered linear kernel.

Equation (35) is especially convenient, because the kernel takes on only two values, $1$ and $-1/(n-1)$. Consequently, precise labels are not needed. This is not the case for unbalanced classes, as $\mathbb{E}\mathbf{y}_i \neq \mathbf{1}_n/n$ and centering of $\mathbf{y}$ doesn't make the normalized vector centered. However, taking the linear kernel gives an almost binary signal,

$$k(\mathbf{y}_i, \mathbf{y}_j) = (\mathbf{y}_i - \mathbf{p})^\top (\mathbf{y}_j - \mathbf{p}) = \mathbb{I}\left[\mathbf{y}_i = \mathbf{y}_j\right] + \sum_k p_k^2 - p_i - p_j$$

$$= \mathbb{I}\left[\mathbf{y}_i = \mathbf{y}_j\right] + \mathcal{O}\left(\frac{1}{n}\right), \tag{36}$$

as long as the probability of each class $k$, $p_k$, is $\mathcal{O}(1/n)$ (i.e., they are roughly balanced). As a result, the teaching signal is nearly binary, and we can compute it without knowing the probability of each class.

## B Derivations of the update rules for plausible kernelized information bottleneck

### B.1 General update rule

Here we derive the gradient of pHSIC in our network, along with its empirical estimate. Our starting point is the observation that because the network is feedforward, the activity in layer $k$ is a deterministic function of the previous layer and the weights: $Z^k = f\left(\mathbf{W}^k, Z^{k-1}\right)$ (this includes both feedforward layers and layers with divisive normalization). Therefore, we can write the expectations of any function $g(Y, Z^k)$ (which need not actually depend on $Y$) in terms of $Z^{k-1}$,

$$\mathbb{E}_{\mathbf{y}\mathbf{z}^k} g(\mathbf{y}, \mathbf{z}^k) = \mathbb{E}_{\mathbf{y}\mathbf{z}^{k-1}} g\left(\mathbf{y}, f\left(\mathbf{W}^k, \mathbf{z}^{k-1}\right)\right). \tag{37}$$

As a result, we can write the gradients of pHSIC (assuming we can exchange the order of differentiation and expectation, and the function is differentiable at $\mathbf{W}^k$) as

$$\frac{d\,\mathrm{pHSIC}\left(Y,\,Z^k\right)}{d\,\mathbf{W}^k} = \left(\mathbb{E}_{\mathbf{y}_1 \mathbf{z}_1^{k-1}} \mathbb{E}_{\mathbf{y}_2 \mathbf{z}_2^{k-1}} - \mathbb{E}_{\mathbf{y}_1} \mathbb{E}_{\mathbf{z}_1^{k-1}} \mathbb{E}_{\mathbf{y}_2} \mathbb{E}_{\mathbf{z}_2^{k-1}}\right)$$
$$k(\mathbf{y}_1, \mathbf{y}_2) \frac{d\,k\left(f\left(\mathbf{W}^k, \mathbf{z}_1^{k-1}\right), f\left(\mathbf{W}^k, \mathbf{z}_2^{k-1}\right)\right)}{d\,\mathbf{W}^k}, \tag{38a}$$

$$\frac{d\,\mathrm{pHSIC}\left(Z^k,\,Z^k\right)}{d\,\mathbf{W}^k} = 2\,\mathbb{E}_{\mathbf{z}_1^{k-1}} \mathbb{E}_{\mathbf{z}_2^{k-1}} k(\mathbf{z}_1^k, \mathbf{z}_2^k) \frac{d\,k\left(f\left(\mathbf{W}^k, \mathbf{z}_1^{k-1}\right), f\left(\mathbf{W}^k, \mathbf{z}_2^{k-1}\right)\right)}{d\,\mathbf{W}^k}$$
$$- 2\left(\mathbb{E}_{\mathbf{z}_1^{k-1}} \mathbb{E}_{\mathbf{z}_2^{k-1}} k(\mathbf{z}_1^k, \mathbf{z}_2^{k-1})\right) \left(\mathbb{E}_{\mathbf{z}_1^{k-1}} \mathbb{E}_{\mathbf{z}_2^{k-1}} \frac{d\,k\left(f\left(\mathbf{W}^k, \mathbf{z}_1^{k-1}\right), f\left(\mathbf{W}^k, \mathbf{z}_2^{k-1}\right)\right)}{d\,\mathbf{W}^k}\right). \tag{38b}$$

To compute these derivates from data, we take empirical averages (see Eq. (34)),

$$
\frac{d\left(\widehat{\mathrm{pHSIC}}(Z^k, Z^k) - \gamma\,\widehat{\mathrm{pHSIC}}(Y, Z^k)\right)}{d\,\mathbf{W}^k} =
$$
$$
2\frac{1}{m^2}\sum_{ij} k(\mathbf{z}_i^k, \mathbf{z}_j^k)\frac{d\,k(\mathbf{z}_i^k, \mathbf{z}_j^k)}{d\,\mathbf{W}^k} - 2\frac{1}{m^2}\sum_{ql} k(\mathbf{z}_q^k, \mathbf{z}_l^k)\frac{1}{m^2}\sum_{ij}\frac{d\,k(\mathbf{z}_i^k, \mathbf{z}_j^k)}{d\,\mathbf{W}^k} \tag{39}
$$
$$
-\gamma\frac{1}{m^2}\sum_{ij} k(\mathbf{y}_i, \mathbf{y}_j)\frac{d\,k(\mathbf{z}_i^k, \mathbf{z}_j^k)}{d\,\mathbf{W}^k} + \gamma\frac{1}{m^2}\sum_{ql} k(\mathbf{y}_q, \mathbf{y}_l)\frac{1}{m^2}\sum_{ij}\frac{d\,k(\mathbf{z}_i^k, \mathbf{z}_j^k)}{d\,\mathbf{W}^k}.
$$

Making the definition $\mathring{k}(\mathbf{a}_i, \mathbf{a}_j) = k(\mathbf{a}_i, \mathbf{a}_j) - \sum_{ql} k(\mathbf{a}_q, \mathbf{a}_l)/m^2$, this simplifies to

$$
\frac{d\left(\widehat{\mathrm{pHSIC}}(Z^k, Z^k) - \gamma\,\widehat{\mathrm{pHSIC}}(Y, Z^k)\right)}{d\,\mathbf{W}^k}
$$
$$
= \frac{1}{m^2}\sum_{ij}\left(2\,\mathring{k}(\mathbf{z}_i^k, \mathbf{z}_j^k) - \gamma\,\mathring{k}(\mathbf{y}_i, \mathbf{y}_j)\right)\frac{d\,k(\mathbf{z}_i^k, \mathbf{z}_j^k)}{d\,\mathbf{W}^k}. \tag{40}
$$

The key quantity in the above expressions is the derivative of the kernel with respect to the weights. Below we compute those for the Gaussian and cosine similarity kernels, and explain why the cosine similarity update is implausible.

## B.2 Gaussian kernel

For the Gaussian kernel, the derivative with respect to a single weight is

$$
\frac{d\,k(\mathbf{z}_i^k, \mathbf{z}_j^k)}{d\,W_{nm}^k} = \frac{d}{d\,W_{nm}^k}\exp\left(-\frac{1}{2\sigma^2}\left\|\mathbf{z}_i^k - \mathbf{z}_j^k\right\|^2\right)
$$
$$
= -\frac{k(\mathbf{z}_i^k, \mathbf{z}_j^k)}{\sigma^2}\left(z_{n,i}^k - z_{n,j}^k\right)\frac{d\left(z_{n,i}^k - z_{n,j}^k\right)}{d\,W_{nm}^k} \tag{41}
$$
$$
= -\frac{k(\mathbf{z}_i^k, \mathbf{z}_j^k)}{\sigma^2}\left(z_{n,i}^k - z_{n,j}^k\right)\left(f'\left(\mathbf{W}^k\mathbf{z}_i^{k-1}\right)_n z_{m,i}^{k-1} - f'\left(\mathbf{W}^k\mathbf{z}_j^{k-1}\right)_n z_{m,j}^{k-1}\right).
$$

For the linear network $f'(x) = 1$, and so the gradient w.r.t. $\mathbf{W}^k$ becomes an outer product (Eq. (12)).

## B.3 Gaussian kernel with grouping and divisive normalization

For the circuit with grouping and divisive normalization, the kernel is a function of $\mathbf{v}$, not $\mathbf{z}$ (see Eqs. (13), (14) and (15)). This makes the derivative with respect to $\mathbf{z}$ more complicated than the above expression would suggest. Specifically, using Eq. (15) for the kernel with grouping, we have

$$
\frac{d\,k(\mathbf{z}_i^k, \mathbf{z}_j^k)}{d\,W_{\alpha nm}^k} = \frac{d}{d\,W_{\alpha nm}^k}\exp\left(-\frac{1}{2\sigma^2}\left\|\mathbf{v}_i^k - \mathbf{v}_j^k\right\|^2\right)
$$
$$
= -\frac{k(\mathbf{z}_i^k, \mathbf{z}_j^k)}{\sigma^2}\sum_{\alpha'}(v_{\alpha',i}^k - v_{\alpha',j}^k)\frac{d\,(v_{\alpha',i}^k - v_{\alpha',j}^k)}{d\,W_{\alpha nm}^k}, \tag{42}
$$

where the sum over $\alpha'$ appears due to centering (so all $\alpha$ are coupled). Using Eq. (14) to express $\mathbf{v}$ in terms of $\mathbf{u}$, the above derivative is

$$
\frac{d\,v_{\alpha',i}^k}{d\,W_{\alpha nm}^k} = \frac{d\left((u_{\alpha',i}^k)^{1-p} - \frac{1}{c_\alpha^k}\sum_{\alpha''}(u_{\alpha'',i}^k)^{1-p}\right)}{d\,W_{\alpha nm}^k} = \left(\delta_{\alpha\alpha'} - \frac{1}{c_\alpha^k}\right)\frac{d\,(u_{\alpha,i}^k)^{1-p}}{d\,W_{\alpha nm}^k}. \tag{43}
$$

Then using Eq. ([13](#)) to express $\mathbf{u}$ in terms of $\mathbf{z}$, we have

$$
\begin{aligned}
\frac{d\,(u_{\alpha,\,i}^k)^{1-p}}{d\,W_{\alpha nm}^k} &= \frac{1-p}{(u_{\alpha,\,i}^k)^p}\,\frac{d\left(\frac{\delta}{c_\alpha^k}+\frac{1}{c_\alpha^k}\sum_{n'}\left(\overset{\circ}{z}{}_{\alpha n',\,i}^k\right)^2\right)}{d\,W_{\alpha nm}^k}\\[6pt]
&= \frac{1-p}{(u_{\alpha,\,i}^k)^p}\,\frac{2}{c_\alpha^k}\sum_{n'}\overset{\circ}{z}{}_{\alpha n',\,i}^k\,\frac{d\,\overset{\circ}{z}{}_{\alpha n',\,i}^k}{d\,W_{\alpha nm}^k}\\[6pt]
&= \frac{1-p}{(u_{\alpha,\,i}^k)^p}\,\frac{2}{c_\alpha^k}\sum_{n'}\overset{\circ}{z}{}_{\alpha n',\,i}^k\,\frac{d\left(z_{\alpha n',\,i}^k-\frac{1}{c_\alpha^k}\sum_{n''}z_{\alpha n'',\,i}^k\right)}{d\,W_{\alpha nm}^k}\\[6pt]
&= \frac{1-p}{(u_{\alpha,\,i}^k)^p}\,\frac{2}{c_\alpha^k}\sum_{n'}\overset{\circ}{z}{}_{\alpha n',\,i}^k\left(\delta_{n'n}-\frac{1}{c_\alpha^k}\right)\frac{d\,z_{\alpha n,\,i}^k}{d\,W_{\alpha nm}^k}\\[6pt]
&= \frac{1-p}{(u_{\alpha,\,i}^k)^p}\,\frac{2}{c_\alpha^k}\overset{\circ}{z}{}_{\alpha n,\,i}^k\,\frac{d\,z_{\alpha n,\,i}^k}{d\,W_{\alpha nm}^k}\,.
\end{aligned}
\tag{44}
$$

Inserting this expression into Eq. ([43](#)), inserting that into Eq. ([42](#)), and performing a small amount of algebra, we arrive at

$$
\frac{d\,k(\mathbf{z}_i^k,\mathbf{z}_j^k)}{d\,W_{\alpha nm}^k} = -\frac{k(\mathbf{z}_i^k,\mathbf{z}_j^k)}{\sigma^2}\sum_{\alpha'}(v_{\alpha',\,i}^k-v_{\alpha',\,j}^k)\left(\delta_{\alpha\alpha'}-\frac{1}{c_\alpha^k}\right)\frac{2\,(1-p)}{c_\alpha^k}
\tag{45}
$$

$$
\times\left(\frac{\overset{\circ}{z}{}_{\alpha n,\,i}^k}{(u_{\alpha,\,i}^k)^p}\,\frac{d\,z_{\alpha n,\,i}^k}{d\,W_{\alpha nm}^k}-\frac{\overset{\circ}{z}{}_{\alpha n,\,j}^k}{(u_{\alpha,\,j}^k)^p}\,\frac{d\,z_{\alpha n,\,j}^k}{d\,W_{\alpha nm}^k}\right).
\tag{46}
$$

As $v_{\alpha,\,i}^k$ is centered with respect to $\alpha$ and the last term doesn't depend on $\alpha'$, the final expression becomes

$$
\frac{d\,k(\mathbf{z}_i^k,\mathbf{z}_j^k)}{d\,W_{\alpha nm}^k} = -\frac{2\,(1-p)\,k(\mathbf{z}_i^k,\mathbf{z}_j^k)}{\sigma^2 c_\alpha^k}(v_{\alpha,\,i}^k-v_{\alpha,\,j}^k)
\tag{47}
$$

$$
\times\left(\frac{\overset{\circ}{z}{}_{\alpha n,\,i}^k}{(u_{\alpha,\,i}^k)^p}\,f'\left(\mathbf{W}^k\mathbf{z}_i^{k-1}\right)_n z_{m,\,i}^{k-1}-\frac{\overset{\circ}{z}{}_{\alpha n,\,j}^k}{(u_{\alpha,\,j}^k)^p}\,f'\left(\mathbf{W}^k\mathbf{z}_j^{k-1}\right)_n z_{m,\,j}^{k-1}\right).
\tag{48}
$$

## B.4 Cosine similarity kernel

Assuming that $\mathbf{z}^k$ is bounded away from zero (because $\mathbf{z}^k/\|\mathbf{z}^k\|$ is not continuous at 0; adding a smoothing term to the norm would help but won't change the derivation), the derivative of the cosine similarity kernel is

$$
\begin{aligned}
\frac{d\,k(\mathbf{z}_i^k,\mathbf{z}_j^k)}{d\,W_{nm}^k} &= \frac{d}{d\,W_{nm}^k}\frac{(\mathbf{z}_i^k)^\top\mathbf{z}_j^k}{\|\mathbf{z}_i^k\|\,\|\mathbf{z}_j^k\|}\\[6pt]
&= \frac{1}{\|\mathbf{z}_i^k\|\,\|\mathbf{z}_j^k\|}\frac{d\left(z_{n,\,i}^k z_{n,\,j}^k\right)}{d\,W_{nm}^k}-\frac{(\mathbf{z}_i^k)^\top\mathbf{z}_j^k}{\|\mathbf{z}_i^k\|^2\,\|\mathbf{z}_j^k\|^2}\frac{d\,\|\mathbf{z}_i^k\|\,\|\mathbf{z}_j^k\|}{d\,W_{nm}^k}\,.
\end{aligned}
\tag{49}
$$

The first derivative is simple,

$$
\frac{d\left(z_{n,\,i}^k z_{n,\,j}^k\right)}{d\,W_{nm}^k} = z_{n,\,i}^k\,f'\left(\mathbf{W}^k\mathbf{z}_j^{k-1}\right)_n z_{m,\,j}^{k-1}+z_{n,\,j}^k\,f'\left(\mathbf{W}^k\mathbf{z}_i^{k-1}\right)_n z_{m,\,i}^{k-1}\,.
\tag{50}
$$

However, in this form it is hard to interpret, as both terms have the pre-synaptic activity at one point and the post-synaptic activity at the other. We can, though, can re-arrange it into differences in activity,

$$
\begin{aligned}
\frac{d\left(z_{n,\,i}^k z_{n,\,j}^k\right)}{d\,W_{nm}^k} &= \left(z_{n,\,i}^k-z_{n,\,j}^k\right)f'\left(\mathbf{W}^k\mathbf{z}_j^{k-1}\right)_n z_{m,\,j}^{k-1}+z_{n,\,j}^k\,f'\left(\mathbf{W}^k\mathbf{z}_j^{k-1}\right)_n z_{m,\,j}^{k-1}\\[4pt]
&\quad+\left(z_{n,\,j}^k-z_{n,\,i}^k\right)f'\left(\mathbf{W}^k\mathbf{z}_i^{k-1}\right)_n z_{m,\,i}^{k-1}+z_{n,\,i}^k\,f'\left(\mathbf{W}^k\mathbf{z}_i^{k-1}\right)_n z_{m,\,i}^{k-1}\\[4pt]
&= -\left(z_{n,\,i}^k-z_{n,\,j}^k\right)\left(f'\left(\mathbf{W}^k\mathbf{z}_i^{k-1}\right)_n z_{m,\,i}^{k-1}-f'\left(\mathbf{W}^k\mathbf{z}_j^{k-1}\right)_n z_{m,\,j}^{k-1}\right)\\[4pt]
&\quad+z_{n,\,i}^k\,f'\left(\mathbf{W}^k\mathbf{z}_i^{k-1}\right)_n z_{m,\,i}^{k-1}+z_{n,\,j}^k\,f'\left(\mathbf{W}^k\mathbf{z}_j^{k-1}\right)_n z_{m,\,j}^{k-1}\,.
\end{aligned}
\tag{51}
$$

The second one is slightly harder,

$$
\begin{aligned}
\frac{d\,\|\mathbf{z}_i^k\|\,\|\mathbf{z}_j^k\|}{d\,W_{nm}^k} &= \|\mathbf{z}_i^k\|\frac{d\,\sqrt{\sum_{n'}(z_{n',j}^k)^2}}{d\,W_{nm}^k} + \|\mathbf{z}_j^k\|\frac{d\,\sqrt{\sum_{n'}(z_{n',i}^k)^2}}{d\,W_{nm}^k}\\
&= \frac{\|\mathbf{z}_i^k\|}{\|\mathbf{z}_j^k\|}z_{n,j}^k\frac{d\,z_{n,j}^k}{d\,W_{nm}^k} + \frac{\|\mathbf{z}_j^k\|}{\|\mathbf{z}_i^k\|}z_{n,i}^k\frac{d\,z_{n,i}^k}{d\,W_{nm}^k}\\
&= \frac{\|\mathbf{z}_i^k\|}{\|\mathbf{z}_j^k\|}z_{n,j}^k f'\left(\mathbf{W}^k\mathbf{z}_j^{k-1}\right)_n z_{m,j}^{k-1} + \frac{\|\mathbf{z}_j^k\|}{\|\mathbf{z}_i^k\|}z_{n,i}^k f'\left(\mathbf{W}^k\mathbf{z}_i^{k-1}\right)_n z_{m,i}^{k-1}.
\end{aligned}
\tag{52}
$$

Grouping these results together,

$$
\begin{aligned}
\frac{d\,k(\mathbf{z}_i^k,\mathbf{z}_j^k)}{d\,W_{nm}^k} &= -\frac{\left(z_{n,i}^k - z_{n,j}^k\right)\left(f'\left(\mathbf{W}^k\mathbf{z}_i^{k-1}\right)_n z_{m,i}^{k-1} - f'\left(\mathbf{W}^k\mathbf{z}_j^{k-1}\right)_n z_{m,j}^{k-1}\right)}{\|\mathbf{z}_i^k\|\,\|\mathbf{z}_j^k\|}\\
&\quad + \frac{z_{n,i}^k f'\left(\mathbf{W}^k\mathbf{z}_i^{k-1}\right)_n z_{m,i}^{k-1}}{\|\mathbf{z}_i^k\|\,\|\mathbf{z}_j^k\|} + \frac{z_{n,j}^k f'\left(\mathbf{W}^k\mathbf{z}_j^{k-1}\right)_n z_{m,j}^{k-1}}{\|\mathbf{z}_i^k\|\,\|\mathbf{z}_j^k\|}\\
&\quad - \frac{(\mathbf{z}_i^k)^\top\mathbf{z}_j^k}{\|\mathbf{z}_i^k\|^2\,\|\mathbf{z}_j^k\|^2}\frac{\|\mathbf{z}_i^k\|}{\|\mathbf{z}_j^k\|}z_{n,j}^k f'\left(\mathbf{W}^k\mathbf{z}_j^{k-1}\right)_n z_{m,j}^{k-1}\\
&\quad - \frac{(\mathbf{z}_i^k)^\top\mathbf{z}_j^k}{\|\mathbf{z}_i^k\|^2\,\|\mathbf{z}_j^k\|^2}\frac{\|\mathbf{z}_j^k\|}{\|\mathbf{z}_i^k\|}z_{n,i}^k f'\left(\mathbf{W}^k\mathbf{z}_i^{k-1}\right)_n z_{m,i}^{k-1}.
\end{aligned}
\tag{53}
$$

As all terms share the same divisor, we can write the last expression more concisely as

$$
\begin{aligned}
\|\mathbf{z}_i^k\|\,\|\mathbf{z}_j^k\|\frac{d\,k(\mathbf{z}_i^k,\mathbf{z}_j^k)}{d\,W_{nm}^k} &= -\left(z_{n,i}^k - z_{n,j}^k\right)\left(f'\left(\mathbf{W}^k\mathbf{z}_i^{k-1}\right)_n z_{m,i}^{k-1} - f'\left(\mathbf{W}^k\mathbf{z}_j^{k-1}\right)_n z_{m,j}^{k-1}\right)\\
&\quad + \sum_{s=i,j}\left(1 - \frac{(\mathbf{z}_i^k)^\top\mathbf{z}_j^k}{\|\mathbf{z}_s^k\|^2}\right)z_{n,s}^k f'\left(\mathbf{W}^k\mathbf{z}_s^{k-1}\right)_n z_{m,s}^{k-1}.
\end{aligned}
\tag{54}
$$

Considering a linear network for simplicity, the weight update over two points $i$, $j$ (negative of a single term in the sum in Eq. (40)) for the cosine similarity kernel becomes

$$
\begin{aligned}
\Delta W_{nm}^k &= M_{ij}^k\left(z_{n,i}^k - z_{n,j}^k\right)\left(z_{m,i}^{k-1} - z_{m,j}^{k-1}\right) - M_{ij}^k\sum_{s=i,j}\left(1 - \frac{(\mathbf{z}_i^k)^\top\mathbf{z}_j^k}{\|\mathbf{z}_s^k\|^2}\right)z_{n,s}^k z_{m,s}^{k-1},\\
M_{ij}^k &= \frac{1}{\|\mathbf{z}_i^k\|\,\|\mathbf{z}_j^k\|}\left(2\,\mathring{k}(\mathbf{z}_i^k,\mathbf{z}_j^k) - \gamma\,\mathring{k}(\mathbf{y}_i,\mathbf{y}_j)\right).
\end{aligned}
\tag{55}
$$

This rule is biologically implausible (or very hard to implement) for two reasons. First, to compute $M_{ij}^k$ the layer needs to track three signal simultaneously: $(\mathbf{z}_i^k)^\top\mathbf{z}_j^k$, $\|\mathbf{z}_i^k\|$ and $\|\mathbf{z}_j^k\|$ (versus only one for the Gaussian kernel, $\|\mathbf{z}_i^k - \mathbf{z}_j^k\|$). Second, assuming point $i$ comes after $j$, the prefactor of the Hebbian term (the sum in Eq. (55)) for $s = j$ can't be computed until the networks receives point $i$. As a result, the update requires three independent plasticity pathways (one for $M_{ij}^k\left(z_{n,i}^k - z_{n,j}^k\right)\left(z_{m,i}^{k-1} - z_{m,j}^{k-1}\right)$ and two for the sum in Eq. (55)), because each term in Eq. (55) combines the pre- and post-synaptic activity on different timescales and with different third factors.

Adding grouping and divisive normalization to this rule is straightforward: we need to use the grouped response $v_\alpha^k$ (Eq. (14)) in the kernel as $k(\mathbf{z}_i^k,\mathbf{z}_j^k) = (\mathbf{v}_i^k)^\top\mathbf{v}_j^k/(\|\mathbf{v}_i^k\|\,\|\mathbf{v}_j^k\|)$, and repeat the calculation above (divisive normalization will appear here too as it results from differentiating $v_\alpha^k$). As it would only make the circuitry more complicated, we omit the derivation. Note that if we use grouping with $p = 0.5$ but don't introduce divisive normalization, our objective resembles the "sim-bpf" loss in [6] (but it doesn't match it exactly, as we also introduce centering of the kernel and group convolutional layers over multiple channels rather than one).

## B.5 Linear kernel

Derivation of the update for the linear kernel only needs the derivative from Eq. (50), therefore by doing the same calculations we obtain

$$
\begin{aligned}
\frac{d\,k(\mathbf{z}_i^k, \mathbf{z}_j^k)}{d\,W_{nm}^k} &= \frac{d\,(\mathbf{z}_i^k)^\top \mathbf{z}_j^k}{d\,W_{nm}^k} = \frac{d\,\left(z_{n,i}^k z_{n,j}^k\right)}{d\,W_{nm}^k} \\
&= -\left(z_{n,i}^k - z_{n,j}^k\right)\left(f'\left(\mathbf{W}^k \mathbf{z}_i^{k-1}\right)_n z_{m,i}^{k-1} - f'\left(\mathbf{W}^k \mathbf{z}_j^{k-1}\right)_n z_{m,j}^{k-1}\right) \\
&\quad + \sum_{s=i,j} z_{n,s}^k f'\left(\mathbf{W}^k \mathbf{z}_s^{k-1}\right)_n z_{m,s}^{k-1}.
\end{aligned}
\tag{56}
$$

This is much easier to compute than the cosine similarity kernel: it only uses $(\mathbf{z}_i^k)^\top \mathbf{z}_j^k$ in the third factor, and needs two plasticity channels rather than three (as now both terms in the sum use the same third factor). However, we couldn't achieve good performance with this kernel.

## C  Circuitry to implement the update rules

In this section we outline the circuitry needed to compute the Hebbian terms and the third factor for the Gaussian kernel (plain and with grouping and divisive normalization).

### C.1  Hebbian terms

Figure 4: **A.** First scenario of the Hebbian updates for two points: plasticity (proportional to $\Delta z_t$, orange line) happens when the activity (blue line) switches from one data point to another. **B.** Second scenario: plasticity happens when the second point comes in some time after the first one, which uses memorized activity from the first point (dashed green line).

In Section 4.4 we proposed online versions for our update rules. For the plain Gaussian kernel (the discussion below also applies to the version with grouping and divisive normalization), the update is

$$
\Delta \mathbf{W}^k(t) \propto M_{t,t-\Delta t}^k (\mathbf{z}_t^k - \mathbf{z}_{t-\Delta t}^k)(\mathbf{z}_t^{k-1} - \mathbf{z}_{t-\Delta t}^{k-1})^\top,
\tag{57}
$$

where $t - \Delta t$ represents the data point before the one at time $t$. In the main text we suggested that $z_{t-\Delta t}^k$ can be approximated by short-term average of activity in layer $k$.

When $\Delta t$ is small, the change $z_{n,t}^k - z_{n,t-\Delta t}^k$ for a neuron $n$ can be computed by a smoothed temporal derivative, implemented by convolution with a kernel $\kappa$ (see Fig. 4A),

$$
z_{n,t}^k - z_{n,t-\Delta t}^k \approx \Delta z_{n,t}^k \equiv (\kappa * z_n^k)(t); \quad \kappa(t) \propto -(t - c_1)\, e^{-c_2|t-c_1|}\, \Theta(t),
\tag{58}
$$

with $c_1$ and $c_2$ are positive.

If $\Delta t$ is large and potentially variable, the short-term average won't accurately represent the previous point. However, if between trials $z_n^k$ returns to some background activity $\mu_n^k$ (see Fig. 4B), we can still apply these difference-based updates. In this case the neuron needs to memorize the last significant deviation from the background (i.e., it needs to remember $z_{n,t-\Delta t}^k - \mu_{n,t-\Delta t}^k$). This can be done with a one-dimensional nonlinear differential equation with "memory", such as

$$
\dot{\omega}_{n,t} = \left(z_{n,t}^k - \mu_{n,t}^k\right)^3 - \tanh\left(\left|z_{n,t}^k - \mu_{n,t}^k\right|^3\right) \omega_{n,t} - c\,\omega_{n,t}
\tag{59}
$$

with $c$ small. The intuition is as follows: if the neuron is in the background state, the right hand side of Eq. (59) is nearly zero (expect for the leak term $c\,\omega_{n,t}$). Otherwise, the deviation from the mean is

large and the right hand side approaches $(z_{n,t}^k - \mu_{n,t}^k)^3 - (1+c)\,\omega_{n,t}$, as long as $|z^k - \mu_{n,t}^k| \gg 1$ (due to tanh saturation). This quickly erases the previous $z_{n,t-\Delta t}^k$ and memorizes the new one (see Fig. 4B with no leak term). We can therefore compute the difference between the last and the current large deviations by convolving the (real) cube root $(\omega_{n,t}^k)^{1/3}$ with the same kernel as above, leading to

$$\mathbf{z}_{n,t}^k - \mathbf{z}_{n,t-\Delta t}^k \approx \Delta z_{n,t}^k \equiv \left(\kappa * (\omega_n^k)^{1/3}\right)(t). \tag{60}$$

### C.2  3rd factor for the Gaussian kernel

The update equation for the Gaussian kernel (repeating Eq. (12) but in time rather that indices $ij$, and assuming a linear network as it doesn't affect the third factor) is

$$\Delta W_{nm,t}^k \propto M_t^k\,(z_{n,t}^k - z_{n,t-\Delta t}^k)(z_{m,t}^{k-1} - z_{m,t-\Delta t}^{k-1}), \tag{61}$$

$$M_t^k = -\frac{1}{\sigma^2}\left(\gamma\,\mathring{k}(\mathbf{y}_t,\mathbf{y}_{t-\Delta t}) - 2\,\mathring{k}(\mathbf{z}_t^k,\mathbf{z}_{t-\Delta t}^k)\right)k(\mathbf{z}_t^k,\mathbf{z}_{t-\Delta t}^k). \tag{62}$$

We'll assume that information about the labels, $k(\mathbf{y}_t,\mathbf{y}_{t-\Delta t})$, comes from outside the circuit, so to compute the third factor we just need to compute $k(\mathbf{z}_t^k,\mathbf{z}_{t-\Delta t}^k)$ (and then center everything). For the Gaussian kernel, we thus need $\left\|\Delta\mathbf{z}_t^k\right\|^2$, which is given by

$$b_{1,t}^k = \sum_n(\Delta z_{n,t}^k)^2, \tag{63}$$

so that $k(\mathbf{z}_t^k,\mathbf{z}_{t-\Delta t}^k) = \exp(-b_{1,t}^k/(2\sigma^2))$. That gives us the uncentered component of the third factor, denoted $b_{2,t}^k$,

$$b_{2,t}^k = \gamma\,k(\mathbf{y}_t,\mathbf{y}_{t-\Delta t}) - 2\,k(\mathbf{z}_t^k,\mathbf{z}_{t-\Delta t}^k) = \gamma\,k(\mathbf{y}_t,\mathbf{y}_{t-\Delta t}) - 2\,\exp\left(-\frac{1}{2\sigma^2}b_{1,t}^k\right). \tag{64}$$

To compute the mean, so that we may center the third factor, we take an exponentially decaying running average,

$$b_{3,t}^k = \beta\,b_{2,t}^k + (1-\beta)\,b_{3,t}^k \tag{65}$$

with $\beta \in (0,1)$ (so that past points are erased as the weights change). Thus, the third factor becomes

$$M_t^k = -\frac{1}{\sigma^2}\left(b_{2,t}^k - b_{3,t}^k\right)\exp\left(-\frac{1}{2\sigma^2}b_{1,t}^k\right). \tag{66}$$

If we think of $b_{1,t}^k$, $b_{2,t}^k$ and $b_{3,t}^k$ as neurons, the first two need nonlinear dendrites to compute this signal. In addition, $b_{1,t}^k$ should compute $\Delta z_{n,t}^k$ from $z_{n,t}^k$ at the dendritic level.

### C.3  3rd factor for the Gaussian kernel with grouping and divisive normalization

The update for the Gaussian kernel with grouping (repeating Eq. (18) but in time, and assuming a linear network as it doesn't affect the third factor) is

$$\Delta W_{\alpha nm,t}^k \propto M_{\alpha,t}^k\left(r_{\alpha n,t}^k r_{m,t}^{k-1} - r_{\alpha n,t-\Delta t}^k r_{m,t-\Delta t}^{k-1}\right), \tag{67a}$$

$$M_{\alpha,t}^k = M_t^k\,(v_{\alpha,t}^k - v_{\alpha,t-\Delta t}^k), \tag{67b}$$

$$M_t^k = -\frac{1}{\sigma^2}\left(\gamma\,\mathring{k}(\mathbf{y}_t,\mathbf{y}_{t-\Delta t}) - 2\,\mathring{k}(\mathbf{z}_t^k,\mathbf{z}_{t-\Delta t}^k)\right)k(\mathbf{z}_t^k,\mathbf{z}_{t-\Delta t}^k). \tag{67c}$$

The Hebbian term – the term in parentheses in Eq. (67a) – corresponds to the difference of pre- and post-synaptic products, rather than a product of differences: $r_{\alpha n,t}^k r_{m,t}^{k-1} - r_{\alpha n,t-\Delta t}^k r_{m,t-\Delta t}^{k-1} \approx \Delta(r_{\alpha n,t}^k r_{m,t}^{k-1})$. We can compute this difference as before (Eq. (58) or Eq. (60)), although this update rule requires a different interaction of the pre- and post-synaptic activity comparing to the plain Gaussian kernel in Eq. (61).

The third factor is almost the same as for the plain Gaussian kernel, but there are two differences. First, we need to compute the centered normalization $v_{\alpha,t}^k$ (as in Eq. (14)) and its change over time for each group $\alpha$,

$$\tilde{b}_{\alpha,t}^k = \Delta v_{\alpha,t}^k . \tag{68}$$

Second, $M_t^k$ is computed for $\Delta v_{\alpha,t}^k$ rather than $\Delta z_{n,t}^k$, such that (cf. Eq. (63))

$$\tilde{b}_{1,t}^k = \sum_\alpha (\tilde{b}_{\alpha,t}^k)^2 , \tag{69}$$

and $\tilde{b}_{2,t}^k$ (Eq. (64) but with $\tilde{b}_{1,t}^k$) and $\tilde{b}_{3,t}^k$ (Eq. (65) but with $\tilde{b}_{1,t}^k$ and $\tilde{b}_{2,t}^k$) stay the same.

The third factor becomes

$$M_{\alpha,t}^k = -\frac{1}{\sigma^2} \left( \tilde{b}_{2,t}^k - \tilde{b}_{3,t}^k \right) \exp\left( -\frac{1}{2\sigma^2} \tilde{b}_{1,t}^k \right) \tilde{b}_{\alpha,t}^k . \tag{70}$$

Essentially, it is the same third factor computation as for the Gaussian kernel, but with an additional group-specific signal $b_{\alpha,t}^k$.

## D  Experimental details

### D.1  Network architecture

Each hidden layer of the network had its own optimizer, such that weight updates happen during the forward pass.

Each layer had the following structure: linear/convolutional operation $\rightarrow$ batchnorm (if any) $\rightarrow$ nonlinearity $\rightarrow$ pooling (if any) $\rightarrow$ local loss computation (doesn't modify activity) $\rightarrow$ divisive normalization (if any) $\rightarrow$ dropout.

None of the hidden layers had the bias term, but the output layer did. The last layer (or the whole network for backprop) was trained with the cross-entropy loss. Non-pHSIC methods with divisive normalization used the same group arrangement, but did not have grouping in the objectives.

### D.2  Choice of kernels for pHSIC

The gradient over each batch was computed as in Eq. (40).

We used cosine similarity with centered labels (Eq. (35); as all datasets are balanced, we don't need to know the probability of a class to center). The kernels for $\mathbf{z}$ were plain Gaussian (Eq. (26)), Gaussian with grouping and divisive normalization ("grp+div"; Eq. (15) such that the next layer sees $r_{\alpha n}^k \equiv \mathring{z}_{\alpha n}^k/(u_\alpha^k)^p$) and grouping without divisive normalization ("grp"; also Eq. (15) but the next layer sees $z_{\alpha n}^k$), plain cosine similarity (Eq. (25)), and cosine similarity with grouping and with or without divisive normalization ($k(\mathbf{z}_i^k, \mathbf{z}_j^k) = (\mathbf{v}_i^k)^\top \mathbf{v}_j^k/(\|\mathbf{v}_i^k\| \|\mathbf{v}_j^k\|)$ with $\mathbf{v}$ as in Eq. (14)).

### D.3  Objective choice for layer-wise classification

As proposed in [6], each convolutional layer is first passed through an average pooling layer such that the final number of outputs is equal to 2048 (e.g. a layer with 128 channels and 32 by 32 images is pooled with an 8 by 8 kernel with stride $= 8$); the resulting 2048-dimensional vector is transformed into a 10-dimensional vector (for class prediction) by a linear readout layer. Fully connected layers are transformed directly with the corresponding linear readout layer. In the layer-wise classification with feedback alignment, feedback alignment is applied to the readout layer.

### D.4  Pre-processing of datasets

MNIST, fashion-MNIST and Kuzushiji-MNIST images were centered by $0.5$ and normalized by $0.5$.

For CIFAR10, each training image was padded by zeroes from all sides with width 4 (resulting in a 40 by 40 image for each channel) and randomly cropped to the standard size (32 by 32),

then flipped horizontally with probability 0.5, and then centered by $(0.4914, 0.4822, 0.4465)$ (each number corresponds to a channel) and normalized by $(0.247, 0.243, 0.261)$. For validation and test, the images were only centered and normalized.

## D.5  Shared hyperparameters for all experiments

We used the default parameters for AdamW, batchnorm, LReLU and SELU; for grouping without divisive normalization we used $p = 0.5$ to be comparable with the objective in [6]. The rest of the parameters (including the ones below) were tuned on a validation set ($10\%$ of the training set for all datasets).

Weight decay for the local losses was 1e-7, and for the final/backprop it was 1e-6; the learning rates were multiplied by 0.25, with individual schedules described below. For SGD, the momentum was 0.95; for AdamW, $\beta_1 = 0.9$, $\beta_2 = 0.999$, $\epsilon =$1e-8. Batchnorm had momentum of 0.1 and $\epsilon = 1e - 5$, with initial scale $\gamma = 1$ and shift $\beta = 0$. Leaky ReLU had the slope 0.01; $\text{SELU}(x) = \text{scale}(\max(0, x) + \min(0, \alpha(\exp(x) - 1)))$ had $\alpha \approx 1.6733$ and scale $\approx 1.0507$ (precise values were found numerically in [43]; note that dropout for SELU was changed to alpha dropout, as proposed in [43]). All convolutions used 3 by 3 kernel with padding $= 1$ (on each side), stride $= 1$ and dilation $= 1$ and no groups; max pooling layers used 2 by 2 kernels with stride $= 2$, dilation $= 1$ and no padding. Grouping with divisive normalization used $\delta = 1$ and $p = 0.2$ (backprop, pHSIC) or $p = 0.5$ (FA, sign symmetry, layer-wise classification), and $p = 0.5$ without divisive normalization. Gaussian kernels used $\sigma = 5$. The balance parameter was set to $\gamma = 2$.

## D.6  Small network

The dropout for all experiments was 0.01, with LReLU for nonlinearity. The networks were trained for 100 epochs, and the learning rates were multiplied by 0.25 at epochs 50, 75 and 90. The individual parameters, $\eta_f$ for final/backprop initial learning rate, $\eta_l$ for the local initial learning rate, $c^k$ for the number of groups in the objective, are given in Table 3; the final results are given in Table 4 (same as Table 1 but with "grp") and Table 5 (max - min accuracy).

## D.7  Large network

The dropout for all experiments was 0.05, with LReLU for AdamW+batchnorm and SELU for SGD. The networks were trained for 500 epochs, and the learning rates were multiplied by 0.25 at epochs 300, 350, 450 and 475 (and at 100, 200, 250, 275 for backprop with SGD). The individual parameters, $\eta_f$ for final/backprop initial learning rate, $\eta_l$ for the local initial learning rate, $c^k$ for the number of groups in the objective, are given in Table 6 and Table 7. The results are given in Table 8 and Table 9 (mean test accuracy) and Table 10 and Table 11 (max - min accuracy). The batch manhattan method mentioned in Table 7 was proposed in [23]; it is used to stabilize feedback alignment and sign symmetry algorithms by substituting the gradient w.r.t. the loss (i.e. before adding momentum and weight decay) with its sign for each weight update. However, in our experiments it didn't improve performance in most of the cases. Without any normalization, we didn't find a successful set of parameters for the Gaussian kernel with grouping and for the methods with feedback alignment and sign symmetry. In those cases, the training either diverged completely or was stuck at low training and even lower test errors (e.g. around 40% training error for the Gaussian kernel with grouping, and around 80% training error for layer-wise classification with feedback alignment).

## D.8  Difference between pHSIC and HSIC in the large network

While we reported all result for pHSIC, training with HSIC instead did not lead to a significant change in the results (not shown). Moreover, the difference between the two objectives stays small during training, as we illustrate below.

As explained in Appendix A.2, our objectives differs from HSIC only in the first term, $\text{pHSIC}\left(Z^k, Z^k\right)$, due to centering of labels. We trained the 1x wide networks from the previous section (grouping + divisive normalization with SGD, grouping + batchnorm with AdamW) and plotted

$$\frac{\text{pHSIC}\left(Z^k, Z^k\right) - \text{HSIC}\left(Z^k, Z^k\right)}{\text{pHSIC}\left(Z^k, Z^k\right)} \tag{71}$$

as a function of training epoch. We compute this quantity on the training data, but the test data gives the same behavior (not shown).

The results show that for both the cosine similarity and the Gaussian kernel, the relative distance between pHSIC and HSIC (Eq. (71)) stays small in all layers expect the first one, but even there it remains relatively constant when trained on the pHSIC objective with SGD + divisive normalization (Fig. 5) or AdamW + batchnorm (Fig. 6); the same holds when the objective is HSIC (Fig. 7 for SGD + divisive normalization and Fig. 8 for AdamW + batchnorm), although earlier layers have larger values when compared to pHSIC training.

Table 3: Parameters for the 3-layer fully connected net (1024 neurons per layer). Last layer: training of the last layer; cossim: cosine similarity; grp: grouping; div: divisive normalization.

| | backprop | | last layer | | pHSIC: cossim | | | pHSIC: Gaussian | | |
|---|---|---|---|---|---|---|---|---|---|---|
| | | div | | div | | grp | grp+div | | grp | grp+div |
| **MNIST** | | | | | | | | | | |
| $\eta_f$ | 5e-2 | 5e-3 | 5e-2 | 5e-2 | 5e-3 | 5e-3 | 5e-3 | 5e-4 | 5e-4 | 1e-3 |
| $\eta_l$ | | | | | 0.5 | 0.6 | 0.4 | 0.6 | 1.0 | 1.0 |
| $c^k$ | | 16 | | 16 | | 16 | 16 | | 32 | 32 |
| **f-MNIST** | | | | | | | | | | |
| $\eta_f$ | 5e-3 | 5e-3 | 5e-2 | 5e-2 | 5e-3 | 1e-3 | 5e-4 | 5e-4 | 5e-4 | 5e-4 |
| $\eta_l$ | | | | | 1.0 | 0.6 | 1.0 | 0.5 | 1.0 | 1.0 |
| $c^k$ | | 32 | | 32 | | 32 | 32 | | 32 | 32 |
| **K-MNIST** | | | | | | | | | | |
| $\eta_f$ | 5e-2 | 5e-2 | 5e-2 | 5e-2 | 5e-3 | 5e-3 | 5e-4 | 1e-3 | 1e-3 | 1e-3 |
| $\eta_l$ | | | | | 0.6 | 0.4 | 0.4 | 0.6 | 1.0 | 1.0 |
| $c^k$ | | 32 | | 16 | | 16 | 16 | | 32 | 32 |
| **CIFAR10** | | | | | | | | | | |
| $\eta_f$ | 5e-3 | 5e-3 | 5e-2 | 1e-2 | 1e-3 | 5e-3 | 5e-3 | 5e-3 | 5e-4 | 1e-3 |
| $\eta_l$ | | 32 | | 32 | 1.0 | 0.4 | 0.1 | 0.1 | 0.6 | 1.0 |
| $c^k$ | | | | | | 32 | 32 | | 32 | 32 |

Table 4: Mean test accuracy over 5 random seeds for a 3-layer fully connected net (1024 neurons per layer). Last layer: training of the last layer; cossim: cosine similarity; grp: grouping; div: divisive normalization.

| | backprop | | last layer | | pHSIC: cossim | | | pHSIC: Gaussian | | |
|---|---|---|---|---|---|---|---|---|---|---|
| | | grp+div | | grp+div | | grp | grp+div | | grp | grp+div |
| MNIST | 98.6 | 98.4 | 92.0 | 95.4 | 94.9 | 95.8 | 96.3 | 94.6 | 98.4 | 98.1 |
| f-MNIST | 90.2 | 90.8 | 83.3 | 85.7 | 86.3 | 88.7 | 88.1 | 86.5 | 88.6 | 88.8 |
| K-MNIST | 93.4 | 93.5 | 71.2 | 78.2 | 80.4 | 86.2 | 87.2 | 80.2 | 92.7 | 91.1 |
| CIFAR10 | 60.0 | 60.3 | 39.2 | 38.0 | 51.1 | 52.5 | 47.6 | 41.4 | 48.4 | 46.4 |

Table 5: Same as Table 4, but max minus min test accuracy over 5 random seeds.

| | backprop | | last layer | | pHSIC: cossim | | | pHSIC: Gaussian | | |
|---|---|---|---|---|---|---|---|---|---|---|
| | | grp+div | | grp+div | | grp | grp+div | | grp | grp+div |
| MNIST | 0.2 | 0.1 | 0.3 | 0.3 | 1.4 | 0.5 | 0.6 | 0.2 | 0.3 | 0.2 |
| f-MNIST | 0.2 | 0.4 | 0.3 | 0.2 | 0.6 | 1.1 | 0.3 | 0.2 | 0.6 | 0.2 |
| K-MNIST | 0.3 | 0.3 | 1.1 | 0.8 | 1.0 | 1.0 | 0.9 | 1.0 | 0.4 | 1.2 |
| CIFAR10 | 0.6 | 0.9 | 1.2 | 1.4 | 1.4 | 2.0 | 1.4 | 0.5 | 1.0 | 0.6 |

Table 6: Parameters for the 7-layer conv nets (CIFAR10; 1x and 2x wide). Cossim: cosine similarity; divnorm: divisive normalization; bn: batchnorm. Empty entries: experiments for which we didn't find a satisfying set of parameters due to instabilities in the methods.

| | backprop | | pHSIC: cossim | | pHSIC: Gaussian | |
|---|---|---|---|---|---|---|
| | | div | grp | grp+div | grp | grp+div |
| **1x wide net + SGD** | | | | | | |
| $\eta_f$ | 5e-3 | 6e-3 | 5e-5 | 5e-4 | | 1e-4 |
| $\eta_l$ | | | 3e-2 | 0.5 | | 0.4 |
| $c^k$ | | 64 | 32 | 64 | | 64 |
| **2x wide net + SGD** | | | | | | |
| $\eta_f$ | 6e-3 | 6e-3 | 5e-5 | 5e-4 | | 1e-4 |
| $\eta_l$ | | | 3e-2 | 0.5 | | 0.4 |
| $c^k$ | | 64 | 32 | 64 | | 64 |
| **1x wide net + AdamW + batchnorm** | | | | | | |
| $\eta_f$ | 5e-3 | 5e-3 | 5e-4 | 5e-4 | 5e-4 | 5e-4 |
| $\eta_l$ | | | 5e-4 | 5e-4 | 5e-3 | 1e-2 |
| $c^k$ | | 64 | 32 | 64 | 64 | 64 |
| **2x wide net + AdamW + batchnorm** | | | | | | |
| $\eta_f$ | 5e-3 | 5e-3 | 5e-4 | 5e-4 | 5e-4 | 5e-4 |
| $\eta_l$ | | | 5e-4 | 5e-4 | 5e-3 | 5e-3 |
| $c^k$ | | 64 | 128 | 64 | 128 | 128 |

Figure 5: Training of 1x networks with pHSIC, SGD and divisive normalization. Y-axis represents $\left(\mathrm{pHSIC}(Z^k, Z^k) - \mathrm{HSIC}(Z^k, Z^k)\right)/\mathrm{pHSIC}(Z^k, Z^k)$. **A.** Cosine similarity kernel **B.** Gaussian kernel.

Figure 6: Training of 1x networks with pHSIC, AdamW and batchnorm. Y-axis represents $\left(\mathrm{pHSIC}(Z^k, Z^k) - \mathrm{HSIC}(Z^k, Z^k)\right)/\mathrm{pHSIC}(Z^k, Z^k)$. **A.** Cosine similarity kernel **B.** Gaussian kernel.

Table 7: Parameters for the 7-layer conv nets (CIFAR10; 1x and 2x wide). FA: feedback alignment; sign sym.: sign symmetry; layer class.: layer-wise classification; divnorm: divisive normalization; bn: batchnorm. Empty entries: experiments for which we didn't find a satisfying set of parameters due to instabilities in the methods.

| | FA | sign sym. | layer class. | |
|---|---|---|---|---|
| | | | | +FA |
| 1x wide net + SGD | | | | |
| $\eta_f$ | | | 5e-3 | |
| $\eta_l$ | | | 1e-3 | |
| 2x wide net + SGD | | | | |
| $\eta_f$ | | | 5e-3 | |
| $\eta_l$ | | | 1e-3 | |
| 1x wide net + SGD + divnorm | | | | |
| $\eta_f$ | 1e-3 | 5e-4 | 5e-3 | 5e-3 |
| $\eta_l$ | | | 5e-3 | 5e-3 |
| $c^k$ | 64 | 64 | 64 | 64 |
| batch manhattan | | + | | |
| 2x wide net + SGD + divnorm | | | | |
| $\eta_f$ | 5e-4 | 5e-4 | 5e-3 | 5e-3 |
| $\eta_l$ | | | 5e-3 | 5e-3 |
| $c^k$ | 128 | 128 | 64 | 128 |
| batch manhattan | | + | | |
| 1x wide net + AdamW + bn | | | | |
| $\eta_f$ | 5e-4 | 5e-4 | 5e-3 | 5e-3 |
| $\eta_l$ | | | 5e-4 | 1e-3 |
| 2x wide net + AdamW + bn | | | | |
| $\eta_f$ | 5e-4 | 5e-4 | 5e-3 | 5e-3 |
| $\eta_l$ | | | 5e-4 | 5e-4 |

Table 8: Mean test accuracy on CIFAR10 over 5 runs for a 7-layer conv nets (1x and 2x wide). Cossim: cosine similarity; divnorm: divisive normalization; bn: batchnorm. Empty entries: experiments for which we didn't find a satisfying set of parameters due to instabilities in the methods.

| | backprop | | pHSIC: cossim | | pHSIC: Gaussian | |
|---|---|---|---|---|---|---|
| | | div | grp | grp+div | grp | grp+div |
| 1x wide net + SGD | 91.0 | 91.0 | 88.8 | 89.8 | | 86.2 |
| 2x wide net + SGD | 91.9 | 90.9 | 89.4 | 91.3 | | 90.4 |
| 1x wide net + AdamW + batchnorm | 94.1 | 94.3 | 91.3 | 90.1 | 89.9 | 89.4 |
| 2x wide net + AdamW + batchnorm | 94.3 | 94.5 | 91.9 | 91.0 | 91.0 | 91.2 |

Table 9: Mean test accuracy on CIFAR10 over 5 runs for a 7-layer conv nets (1x and 2x wide). FA: feedback alignment; sign sym.: sign symmetry; layer class.: layer-wise classification; divnorm: divisive normalization; bn: batchnorm. Empty entries: experiments for which we didn't find a satisfying set of parameters due to instabilities in the methods.

| | FA | sign sym. | layer class. | |
|---|---|---|---|---|
| | | | | +FA |
| 1x wide net + SGD | | | 90.0 | |
| 2x wide net + SGD | | | 90.3 | |
| 1x wide net + SGD + divnorm | 80.4 | 89.5 | 90.5 | 81.0 |
| 2x wide net + SGD + divnorm | 80.6 | 91.3 | 91.3 | 81.2 |
| 1x wide net + AdamW + bn | 82.4 | 93.6 | 92.1 | 90.3 |
| 2x wide net + AdamW + bn | 81.6 | 93.9 | 92.1 | 91.1 |

Table 10: Same as Table 8, but max minus min test accuracy over 5 random seeds.

| | backprop | pHSIC: cossim | | | pHSIC: Gaussian | |
|---|---|---|---|---|---|---|
| | | div | grp | grp+div | grp | grp+div |
| 1x wide net + SGD | 0.4 | 0.4 | 0.7 | 0.7 | | 0.9 |
| 2x wide net + SGD | 0.3 | 0.3 | 2.4 | 0.2 | | 0.5 |
| 1x wide net + AdamW + batchnorm | 0.3 | 0.4 | 0.2 | 0.5 | 0.3 | 0.5 |
| 2x wide net + AdamW + batchnorm | 0.5 | 0.3 | 0.3 | 0.3 | 0.4 | 0.5 |

Table 11: Same as Table 9, but max minus min test accuracy over 5 random seeds. *The large deviation is due to one experiment with about 85% accuracy.

| | FA | sign sym. | layer class. | +FA |
|---|---|---|---|---|
| 1x wide net + SGD | | | 0.4 | |
| 2x wide net + SGD | | | 0.1 | |
| 1x wide net + SGD + divnorm | 1.3 | 5.5* | 0.4 | 1.1 |
| 2x wide net + SGD + divnorm | 0.9 | 0.4 | 0.1 | 0.9 |
| 1x wide net + AdamW + bn | 0.4 | 0.3 | 0.6 | 0.5 |
| 2x wide net + AdamW + bn | 0.9 | 0.4 | 0.1 | 0.4 |

Figure 7: Training of 1x networks with HSIC, SGD and divisive normalization. Y-axis represents $\left(\mathrm{pHSIC}(Z^k, Z^k) - \mathrm{HSIC}(Z^k, Z^k)\right)/\mathrm{pHSIC}(Z^k, Z^k)$. **A.** Cosine similarity kernel **B.** Gaussian kernel.

Figure 8: Training of 1x networks with HSIC, AdamW and batchnorm. Y-axis represents $\left(\mathrm{pHSIC}(Z^k, Z^k) - \mathrm{HSIC}(Z^k, Z^k)\right)/\mathrm{pHSIC}(Z^k, Z^k)$. **A.** Cosine similarity kernel **B.** Gaussian kernel.