[Reviews · NeurIPS 2020]

Review 1

Summary and Contributions: The paper proposes a layer-wise objective based on the concept of the information bottleneck for backpropagation-free supervised learning in deep neural networks. Moreover, the authors show how their learning rule can be related to a possibly biologically plausible learning rule. Finally, the authors analyze the effectiveness of their proposed learning rule on MNIST and CIFAR10 where they show good performance.

Strengths: The authors' back-propagation-free kernel approach to representation learning in deep neural networks is laudable. It holds the potential not only to discover new learning algorithms that do not suffer from backward locking but also to generate a deeper understanding of the mechanistic underpinnings of representation learning in deep neural architectures.

Weaknesses: The main weakness is that the authors did not empirically verify the various approximations that go into the learning rule, other than showing that the learning rule works well in a final training setup. To gain a better understanding of well the chosen approximations hold (e.g. Eq. (6)), would be helpful to better understand why the proposed method works. Moreover, a direct comparison of the proposed learning algorithm to similar previous work, e.g., Nøkland and Eidnes, (2019), using the same network and dataset is missing.

Correctness: The analytical derivations seem sound. Merely for some of the approximations many readers won't have a good intuition as to how much of an error they introduce.

Clarity: Overall, the paper is well written and quite clear. However, the abstract could be reworked by reducing the amount of slang, e.g., "learn to squeeze as much information as possible," and emphasizing the premise of the present study.

Relation to Prior Work: Yes, previous work is mentioned and cited appropriately. The only suggestion for improvement would be to discuss other greedy layer-wise training approaches.

Reproducibility: Yes

Additional Feedback: Consider discussing (Mostafa et al., 2018) as the study underlying (Nøkland and Eidnes, 2019). For central approximations such as Eq. (6) it would be great to provide empirical support for how well the approximation is justified. For instance, sample pairs for both the lhs and rhs and put on a scatter plot. This point is to answer whether this is indeed a good approximation or whether the replacement simply gives rise to a learning rule that works. pg3.98: "But if knew how" ... insert "we" Doesn't Eq. (11) break the plausible online character? The steps from Eq. (13) to (16) were a tad fast for me. It wasn't clear whether this temporal variant is verified empirically in the experiments section or whether it is just an addendum to the theory. I suggest also mentioning the results obtained with Adam in the main manuscript instead of postponing them to the appendix. Finally, the notion of "grouping" was not clear to me and not explained in the text. However, since grouping is applied generously in the experiments section, this term should be explained more clearly in the main text. # UPDATE: Thanks for the clarifications and for addressing my suggestions.


Review 2

Summary and Contributions: Building on the information bottleneck principle and kernel methods the authors propose a layer-wise learning rule that has 3-factor structure: pre(post)-synaptic activities and a 3-rd factor, and can be applied to training artificial neural networks.

Strengths: This is a very interesting paper that contains some new ideas as well as tricks that improve performance. Experimental evaluations demonstrate that the proposed learning rule can learn useful representations in hidden layers for ANNs.

Weaknesses: I think this paper would benefit from a more comprehensive connection to existing ideas in the field of biologically-plausible learning. Specifically: 1. There is a body of literature on biologically plausible methods for training feedforward ANNs with labels, see [R1] and [R2] for example. 2. Biologically-plausible learning rules that do not require label information have been studied in [R3] for shallow networks, and in [R4] for training multiple hidden layers of representations. The latter work, although targeting similarity search as a downstream task and not classification, also extensively uses divisive normalization for images. Similarly to this work, this seems to be important for achieving high accuracy (precision). 3. The empirical evaluations (Table 1) need to be compared with previously published results in the biologically-plausible settings. For instance, [R3] reports accuracy on fully connected network for MNIST better than 98.5%. Refs [1,3] report slightly better than 50% accuracy for fully connected architectures on CIFAR-10. Refs: R1. https://arxiv.org/abs/1412.7525 R2. https://www.frontiersin.org/articles/10.3389/fncom.2017.00024/full R3. https://www.pnas.org/content/116/16/7723 R4. https://arxiv.org/abs/2001.04907

Correctness: I don’t understand the approximation in Eq 6 leading to the pHSIC learning rule. Could the authors please provide some intuition/computational reason on why this approximation is a meaningful thing to do? Same question pertaining to approximations leading to equation 8 from equation 7. -------Post Rebuttal------- Thanks for the clarifications! Although I agree with some of the reviewer’s comments that this paper is not perfectly written, I still think that the originality of the proposed learning rule outweighs problems with the presentation (that need to be fixed). I am inclined to keep my initial scores and still vote the acceptance. I hope that the authors take seriously the issues raised during the review: adjust the claims on biological plausibility of the proposed method, and add the missing references.

Clarity: Clarity can be improved. Certain technical details are missing. For example I had to read Ref [6] from the paper to understand the details of the convolutional architecture implementation. It would be nice if the authors made it self-complete by specifying sizes of convolutional and pooling filters, strides, etc. I am not sure about the overall sign convention in equations 12, but there is a misprint in hebbian/anti-hebbian learning in one of the two instances in lines 133-135.

Relation to Prior Work: As I explained above, the paper can be improved by more comprehensively connecting and comparing this current work with previously published results.

Reproducibility: Yes

Additional Feedback:


Review 3

Summary and Contributions: This paper proposes a new biologically plausible learning rule based on the Hilbert-Schmidt Independence Criterion. The new rule improves on past biological learning rules through its unified training/inference stages and the requirement for a large amount of data. The rule is then demoed on MNIST and CIFAR-10 datasets.

Strengths: The rule proposed does indeed appear biologically plausible. Moreover, the rule depends on a global scalar feedback term and pairwise cross-layer activations - both of which can be interpreted as biological substrates implicated in learning and plasticity.

Weaknesses: The main weaknesses of the paper are 1) the lack of comparison to other biologically plausible rules, 2) the simplicity of the datasets the rule is tested on, and 3) several unsubstantiated points from the abstract. (1) While the paper compares the performance of their proposed rule with various kernels across 4 small datasets, the paper lacked a thorough comparison to other state-of-the-art biologically plausible learning rules. (2) Even on the relatively simple datasets of MNIST and CIFAR10, the proposed methods significantly underperform small networks trained with SGD. Moreover, the most biologically plausible of the proposed kernels - the Gaussian kernel - performs the worst of all methods displayed. This suggests that the proposed rule does not scale to larger problems. (3) The abstract suggests the new rule is novel due to its superior performance in the absence of large amounts of data and the absence of an inference/learning bifurcation. Neither of these claims were adequately supported by the data presented. For one, evidence was not supplied suggesting these new methods performed better than competing biologically plausible rules when trained on little data. Second, the inference/learning bifurcation contained in the weight updates suggested here did not appear any different from previously proposed methods.

Correctness: The derivation of the biologically plausible rule appeared sound and correct.

Clarity: This paper is well written, however I would have appreciated more comparison with other competing methods.

Relation to Prior Work: The authors contrast the new proposed learning rule with previous rules such as feedback alignment on two primary topics, however these two points could benefit from further elaboration.

Reproducibility: Yes

Additional Feedback:


Review 4

Summary and Contributions: A 3-factor Hebbian learning rule is proposed that provides a biologically plausible alternative to BP for training feedforward artificial neural networks.

Strengths: The derivation of the rule integrates a number of clever ideas. The performance is tested for CIFAR10 on CNNs. It achieves there a performance very close to BP.

Weaknesses: The stated goal is to provide a biologically plausible alternative to BP. Unfortunately "biologically plausible" is a rather subjective term. For example, one could point to several aspects of the resulting paradigm which someone could find to be not biologically plausible, such as -feedforward rather than recurrent NNs -CNNs (weight sharing, max-pooling) -use of the SELU activation function, that assumes both positive and negative values -transfer of the supervised learning paradigm from ML, rather than taking into account that brains are likely to combine a lot of unsupervised and self-supervised learning with occasional labels from a teacher Also, the resulting Hebbian plasticity rule looks really complicated, and it is not clear how it can be related to experimental data for synaptic plasticity.

Correctness: As far as I can see, yes.

Clarity: The writing is rather technical, but pretty clear.

Relation to Prior Work: I am missing a discussion of Broadcast Alignment (Lillicrap et al) and methods proposed by Bengio et al.as biologically plausible alternatives to BP, and especially a performance comparison with these. The new method is also not compared with other approaches based on Information Bottleneck or kernels.

Reproducibility: Yes

Additional Feedback:

[Author Response · NeurIPS 2020]

**R1-5. Comparison with previous work.** In the current version, we compare our work with layer-wise objectives (Sec. 1; Sec. 2-4 compare with [Nøkland, 2019]). Comparison with biologically plausible work in Sec. 5 was limited to variations of Feedback Alignment (FA) [Lillicrap, 2016; Moskovitz, 2018; Akrout, 2019]. We'll add more details, including experimental results, which are consistent with [Moskovitz, 2018]: FA performs significantly worse than backprop (80% on the 1x net from Sec. 4.3), sign-symmetry [Liao, 2016] is comparable with our methods (91% with batchnorm; our methods achieve 90-91%) but requires signs of the feedforward weights. Note that all those algorithms need a backward pass and a dedicated circuitry to propagate it, and don't perform better than backprop. We'll reference target prop [Bengio, 2014; Lee, 2015] and equilibrium prop [Scellier, 2017], but they don't seem to scale to large networks/hard problems ([Bartunov, 2018] report that target prop performs worse than FA). We'll extend the comparison with layer-wise objectives (with e.g. [Mostafa, 2018; Krotov, 2019]). We do compare our work with [Nøkland, 2019], using the same architecture in Sec. 4.3 and a slightly modified rule: cosine similarity + grouping corresponds to sim-bpf in [Nøkland, 2019], but achieves better results (88.8% w/o batchnorm and 91.3% with it (Appendix D.6) vs. 86.6% with sim-bpf and 91% when using labels in each layer, both with batchnorm). We will make the comparison more explicit (also see Appendix B.4 for why cosine similarity is implausible). Regarding Information Bottleneck (IB) and kernel approaches, we are not aware of any papers claiming good performance with IB on hard tasks; the only kernel approach to layer-wise training we know of is HSIC bottleneck, which doesn't scale well (<60% accuracy on CIFAR10 in a 5-block ResNet [Ma, 2019]; our experiments weren't successful either); both approaches are discussed in Sec. 1-2.

**R1-2. HSIC/pHSIC approximation (Eqs. 6-7).** We explain in Appendix A.2 why it holds mathematically: $\text{pHSIC}(A, A) - \text{HSIC}(A, A) = 2\mathbb{V}\text{ar}_{a_1}[\mathbb{E}_{a_2} k(a_1, a_2)]$ (small for low variance of "similarity to the mean" $\mathbb{E}_{a_2} k(a_1, a_2)$).

Empirically, the approximation is tight: the figure shows that the relative difference between pHSIC and HSIC stays small throughout training with the pHSIC objective (1x conv net from Sec. 4.3, Gaussian kernel with grouping and divnorm). Training with HSIC instead generally increased the test accuracy by 1-2%. We'll add those results and provide more intuition in the main text.

**R1. Eq. 11** breaks the online character, but we explained how it can be computed online (as an exponential running average) in Appendix C.1 (we'll clarify this in the main text). **Sec 3.3 (eq 13 to 16):** we added it because our rule is unusual as it takes two points, raising question of how it can be plausibly computed. However, we need a batch version to use GPUs. **Adam/batchnorm:** we'll move it to the main text. **Grouping:** we'll add more clarification. Briefly, instead of computing kernels over all neurons (say 100), we arrange them into groups (say 4, with 25 neurons per group) with a single value representing each group and compute kernels over those groups (over 4-dim vectors).

**R2. Tab. 1:** we appreciate the references for Tab. 1 and will improve the comparisons, but it's worth noting the main result is Tab. 2 (larger networks, much better performance). **Eqs. 7-8:** we're sorry for the confusion, Eq. 8 had a typo (RHS was HSIC(A,A), not pHSIC) corrected in the appendix (in the first lines). **Conv parameters:** we overlooked this and will add them to the paper. **Eq. 12** is correct (cf. Eq. 10; derivation in Appendix B.2).

**R4. Performance:** while SGD outperforms our rules in small networks (which still work well with grouping+divnorm on MNIST datasets), we show it is due to the size of those networks – in Sec. 4.3 our rules nearly match SGD performance in large nets (Tab. 2: 90-91% for our methods vs. 91-92% for SGD in the 2x net). **Plausibility/performance:** the Gaussian kernel with grouping and divisive normalization performs much better than the plain one without losing plausibility. As seen in Eq. 22, the update stays Hebbian and the third factor receives a single additional multiplier, computed from the the normalizer in each group. **Large amounts of data:** we're sorry for misunderstanding, but in the abstract we mentioned large amounts of *labeled* data, which we did address with a weakly supervised rule that only requires a binary similarity signal. **Inference/training bifurcation:** our rule doesn't have a backward pass at all, as every layer performs Hebbian learning independently (without signals from the upstream layers).

**R5.** We agree "biological plausibility" is subjective, but we did try to address some of your points and we'll elaborate on them in the paper. **Recurrence:** networks in Sec 4.3 are "recurrent" in the sense that they use divisive normalization. While in a deep feedforward network this works as simple division, a more realistic implementation would require a recurrent computation. We don't have other types of recurrence common in the visual stream (e.g. feedback connections), and introducing them would be an interesting future direction. **CNNs** are indeed implausible, and removing weight sharing is a future direction (however, most cited work on the same topic also uses CNNs). **Negative firing rates:** as SELU saturates for negative values, it can be thought of as firing rate relative to the background. **Supervised learning:** we agree the supervised framework is implausible, and that's what we tried to address with our learning rule: instead of label information, it only needs a binary similarity signal for two consecutive points (the output layer uses labels only to measure accuracy). Such a learning signal is reminiscent of self-supervised learning (e.g. [van den Oord, 2018; Löwe, 2019]) and can likely be applied in that setting. **Complicated update rule:** the Gaussian kernel with grouping and divisive normalization (the main focus of the experiments) needs to combine only three locally accessible signals: global modulation, changes in the local activity, and changes in normalization.

[Meta-Review · NeurIPS 2020]

The reviews and scores on this paper were a bit divergent, but in the end, the reviewers agreed that, with the modifications proposed in the rebuttal, the paper presents an interesting model that is technically sound, and thus, appropriate for acceptance. However, the reviewers also all agreed that the authors need to scale back their claims of biological plausibility in the camera ready version. In particular, the following issues were noted in discussion: 1) The algorithm proposed still has a number of biologically implausible components to it, whether it is lack of excitatory recurrence, negative activities, etc. This is fine, but these failings should be recognized as short-comings, not swept away as unproblematic. 2) The proposed learning algorithm is spatially local, but not Hebbian in the typical meaning of the word. Specifically, the algorithm uses correlations across data points, not correlations in real-time (per Hebb's original proposal), and so is not in-line with what most people understand a "Hebbian" algorithm to be. Thus, the learning rule should be called "local", but not "Hebbian". 3) The supposed global third factor actually is a set of layer-by-layer (or group-by-group) calculations involving both activity from the layer and information about labels (the claim that labels aren't needed only works with one-hot outputs, which is also biologically questionable). Moreover, this third factor still must be communicated backwards to the appropriate layers at the appropriate time, i.e. a backward pass is still required. So, again, the references to a "global" signal, or claims that the need for labels or backward passes is eliminated, should be taken out of the paper.